# Mesoporous Carbon-Based Materials for Enhancing the Performance of Lithium-Sulfur Batteries

**DOI:** 10.3390/ijms24087291

**Published:** 2023-04-14

**Authors:** Fangzheng Wang, Yuying Han, Xin Feng, Rui Xu, Ang Li, Tao Wang, Mingming Deng, Cheng Tong, Jing Li, Zidong Wei

**Affiliations:** School of Chemistry and Chemical Engineering, Chongqing University, Daxuecheng South Road 55, Chongqing 401331, China; fzw20250606@163.com (F.W.); yyhan3560@163.com (Y.H.); 15822216050@163.com (R.X.); liangcqu@outlook.com (A.L.); 20201801032@stu.cqu.edu.cn (T.W.); mingdoc@cqu.edu.cn (M.D.); 20136167@cqu.edu.cn (C.T.)

**Keywords:** lithium-sulfur batteries, energy-storage devices, shuttle effect, carbon materials, mesoporous structure

## Abstract

The most promising energy storage devices are lithium-sulfur batteries (LSBs), which offer a high theoretical energy density that is five times greater than that of lithium-ion batteries. However, there are still significant barriers to the commercialization of LSBs, and mesoporous carbon-based materials (MCBMs) have attracted much attention in solving LSBs’ problems, due to their large specific surface area (SSA), high electrical conductivity, and other unique advantages. The synthesis of MCBMs and their applications in the anodes, cathodes, separators, and “two-in-one” hosts of LSBs are reviewed in this study. Most interestingly, we establish a systematic correlation between the structural characteristics of MCBMs and their electrochemical properties, offering recommendations for improving performance by altering the characteristics. Finally, the challenges and opportunities of LSBs under current policies are also clarified. This review provides ideas for the design of cathodes, anodes, and separators for LSBs, which could have a positive impact on the performance enhancement and commercialization of LSBs. The commercialization of high energy density secondary batteries is of great importance for the achievement of carbon neutrality and to meet the world’s expanding energy demand.

## 1. Introduction

Today, more than 86% of the world’s energy consumption is still ruled by traditional fossil fuels (coal, gas, and oil), leading to resource depletion and significant carbon dioxide (CO_2_) emissions, resulting in a serious “greenhouse effect”. As a result, the concept of “carbon neutrality” has been proposed by 125 countries [1]. It is imperative to create new secondary batteries with high energy densities, in order to attain carbon neutrality and, concurrently, to fulfill the rising energy needs of electric vehicles and other electronic gadgets [2]. LSBs are expected to play a significant role in achieving this, due to their outstanding theoretical specific capacity (1672 mah g^−1^) and theoretical energy density (2600 Wh kg^−1^) [3,4,5,6,7]. As shown in Figure 1, LSBs are a reactive cell because their capacity contribution comes from the four-step redox of S. The battery is fully discharged when S_8_ is completely converted to Li_2_S. However, there are still many obstacles in the way of LSB technology advancement. S and Li_2_S are insulators of electrons and ions (the electronic conductivity of sulfur at room temperature is about 10–30 S cm^−1^), which will perform less well in terms of active material usage, rate capability, and overpotential [8,9,10,11]. In addition, Li_2_S_x_ (4 ≤ x ≤ 8) dissolved in electrolyte will shuttle between the anode and cathode, eventually forming insoluble Li_2_S_2_/Li_2_S deposited on the Li anode, resulting in poor performance [12,13,14,15]. Finally, in the reaction of converting S_8_ into Li_2_S, there is a volume expansion of about 80%, which easily leads to the collapse of the electrode material structure and rapid capacity decay [16,17].

Porous materials have drawn a lot of interest because of their distinctive characteristics and diverse range of uses [18,19]. Porous materials are classified as microporous (d < 2 nm), mesoporous (2 < d < 50 nm), and macroporous (d > 50 nm) by the International Union of Pure and Applied Chemistry (IUPAC) [20,21]. Micropores and small mesopores can provide a high surface area, allowing full contact between sulfur and carbon, and providing abundant LiPSs adsorption sites; large mesopores and macropores can be used as buffer vessels for volume changes and as transport channels for Li^+^. However, the regulation and control of micropores and macropores is less developed than that of mesopores, which are more promising for applications relative to the different needs posed by LSB cathodes, anodes, and separators. Mesoporous carbon materials are promising for various applications in LSBs due to their large specific surface area (SSA), good electrochemical stability, superior electrical conductivity, large pore capacity, and adjustable pore size and pore channel [22,23]. These advantages can solve some specific problems in different batteries [24,25,26]. Mesoporous carbon materials have a large SSA that can offer a large number of reaction sites and active absorption sites when used as carriers for catalytic species [27,28]. In many batteries, the total pore volume (TPV) can accommodate more guest materials, and also buffer the volume expansion generated during repeated electrochemical reactions, and the adjustable pore size and channels can accelerate the translocation of ions and intermediate species to improve the reaction rate [29,30,31]. In addition, these appropriately sized mesopores are physically and chemically constrained, avoiding the loss or aggregation of active materials in certain energy storage devices and catalytic reactions [32].

Figure 2 shows some examples of the application of mesoporous carbon-based materials (MCBMs) in anodes, cathodes, separators, and two-in-one hosts of LSBs to boost the working capacity, which has positive implications for the commercialization process of LSBs. In 2009, Li et al. [33] first used CMK-3/S as an optimized cathode for LSBs, and the LSBs’ performance was significantly enhanced. After that, more research focused on MCBM. When MCBM is used as an anode host, its mesoporous structure can better accommodate the volume change caused by lithium stripping/plating during cycling. In addition, the smaller mesopores can inhibit LiPSs migration without affecting Li^+^ transport, thus mitigating the “shuttle effect”. This also significantly reduces the density of LiPSs in the anode region, prevents the corrosion of the Li anode, and avoids the production of a thick sulfide film on Li anode, thus ensuring the stability of the Li anode. Secondly, the high SSA greatly decreases the local current density, thus inhibiting the lithium dendrite formation during the initial nucleation stage. Thirdly, MCBM has high electronic conductivity, which results in a uniform current density distribution and facilitates uniform Li nucleation and growth. The conductive carbon framework can significantly increase the conductivity of S species when MCBM acts as a host. In addition, the MCBM cathode has a large SSA, which allows the loading of copious active sites for sulfur redox reactions. The MCBM can also be used to modify the separators, as the mesoporous carbon layer can work as an additional current collector to enhance conductivity. LiPSs also undergo redox reactions on the conductive mesoporous layer, and, thus, can improve the active material utilization [34]. In recent years, MCBMs with better properties for accommodating Li and S were developed, which combine the advantages mentioned above to further enhance the performance of LSBs.

The capacity, cycling stability and discharge/charge performance of LSBs can all be significantly enhanced because of the advantages of MCBMs that have already been described. This article focuses on the development of MCBMs research in LSBs, covering its synthesis method and application in anodes, cathodes, separators, and two-in-one host optimization. The relationship between the structural properties (TPV, SSA, pore size and ordering) and electrochemical performance of MCBM will be systematically discussed. In this paper, the influence of structure on battery performance is systematically described based on the methodology that structure determines properties. Concurrently, in order to commercialize LSBs and further achieve carbon neutrality to meet the expanding energy demand of the world, this paper provides ideas for the rational design of LSBs, based on the summary of a large number of achievements.

## 2. Synthesis Strategies of MCBM

MCBMs are mainly solid-phase materials in which the main medium is carbon with a large number of mesopores. Ordered mesoporous materials originated in 1992, when Kesge and Beck et al. [49] for the Mobil Company, successfully synthesized the M41S series ordered mesoporous zeolite by using alkyl quaternary ammonium salt cationic surfactant as a template and assembling it with inorganic silicon species under alkaline conditions. The concept of a “template” was introduced for the first time. Due to the limitations of pore size, many pore modification works of microporous carbon materials cannot be carried out, which cannot meet the industrial requirements of LSBs. MCBMs with pore sizes of 2–50 nm break through these limitations and provide sufficient active site for LiPSs’ adsorption and catalysis. There are four general strategies for MCBM’s synthesis, including the soft templates method, the hard templates method, soft/hard combined templates, and the template-free method.

### 2.1. Soft Templates

The soft templates method offers tremendous control over the mesopores architecture, size, and morphology of MCBM, while being simple and adaptable. The amphiphilic molecules that serve as templates in this procedure first self-assemble into micelles with the precursors. Then, heat treatment can be used to remove the amphiphilic molecules in order to create ordered mesopores. The interaction control between the template and the precursor is essential to this process [50].

In 2004, Dai et al. [51] successfully prepared well-organized mesoporous C films by using resorcinol as a carbon source and PS-P4VP as a soft template, which was the beginning of the construction of MCBM by soft template. Subsequently, as shown in Figure 3a, in 2007, Zhao et al. [52] succeeded in synthesizing a series of well-organized mesoporous C materials using PEO–PPO–PPO/PPO–PEO–PEO as a soft template and phenolic resins as the carbon precursor. The structure of mesoporous carbon could be easily altered, allowing for the phase transition from Fd3m to Im3m, P6mm, and, finally, to Ia3d, by merely changing the proportion of each element. As shown in Figure 3b, in 2016, Su et al. [53] employed the soft template F127 to create ordered mesoporous C that was doped with nitrogen using the water co-assembly method alone, with L-lysine or L-arginine as a catalyst for polymerization and a nitrogen dopant. This technique enabled the mass manufacture of mesoporous carbon doped with nitrogen with a large SSA (600 m^2^ g^−1^), while its composition, structure, and morphology could be tuned through the proportion of components to regulate. In 2020, Kang et al. [54] reacted Fe precursors with dopamine, F127, and 1,3,5-trimethylbenzene (TMB) in aqueous/ethanolic solutions by ligand-assisted polymerization assembly to obtain polymer composites. Here, dopamine was the carbon source, F127 was the soft template, and TMB was the pore reamer, followed by a process of thermal annealing to generate mesoporous C spheres with abundant Fe single atoms, which had a well-defined mesoporous structure loaded with highly dispersed Fe single-atom catalysts in a graphitized carbon skeleton. Meso-Fe-N-C had excellent battery performance because of its open mesoporous structure and high SSA, which enabled it to maximize the usage of active Fe-N_x_ sites.

The self-assembly mechanism of soft template has also been widely studied. Zhao et al. proposed a single-micelle-induced assembly mechanism, which demonstrated that surfactants first generate single micelles or aggregate micelles, then micelles assemble with oligomers or precursors to generate organized mesoporous structures on the interface. A series of structures were prepared by the single micelle assembly method, and the synthesis of MCBM from 0D to 3D was achieved [58,59]. These mechanisms provide the theoretical guidance for the design and synthesis of MCBM.

### 2.2. Hard Templates

Another significant method for synthesizing MCBM is the hard templates method. Most hard templates have mesopores and walls in the range of 2–50 nm, meaning that the materials copied from them also have pores in this range. The hard templates method involves the assembly and growth of precursors in a limited region, using porous solid materials with stable mesoporous structure as templates. Compared with the soft templates method, it is not necessary to strictly regulate the hydrolysis and condensation of the precursors, which is particularly suitable for the synthesis of some MCBM, the sol–gel process of which is difficult to control. At the same time, due to the limited domain of rigid channels, precursor crystallization and growth at a reasonably high temperature accelerated the graphitization of MCBM. This method further expands the framework composition of MCBM. The hard templates method involves four main steps, which are template synthesis, precursor loading, precursor conversion, and template removal.

In 1999, Ryoo et al. [60,61] repared CMK-1, a mesoporous C material with the opposite structure to that of the hard film plate MCM-48, for the first time. Later, Hyeo et al. [62] used MCM-48 aluminosilicate with a 3D channel as a hard template, formaldehyde and phenol as a carbon source to prepare a new type of mesoporous C with high SSA. Subsequently, the hard templates method used for the synthesis of MCBM had been rapidly developed. In 2000, Ryoo et al. [63] reported the synthesis of well-organized mesoporous C (CMK-3) by using SBA-1510 as a hard template and sucrose as a carbon precursor. The structural symmetry of the hard template was preserved in ordered mesoporous carbon, which was the first time this had been achieved. CMK-3 had moderately homogenous mesopores (d = 4.5 nm), a large SSA (1520 m^2^ g^−1^), and a large TPV of 1.3 cm^3^ g^−1^. As shown in Figure 4b, Lu et al. [64] combined octadecyltrimethylammonium bromide with 1,4-bis(triethoxysilyl)benzene, then carbonized to create mesoporous carbon/silica nanocomposites. The silica was removed to create mesoporous carbon materials, which had a large SSA (860 m^2^ g^−1^) and a TPV of 0.7 cm^3^ g^−1^.

As shown in Figure 4c, in 2015, Huang et al. [65] used mesoporous SiO_2_ as a hard template and poly-furfuryl alcohol (PFA) as a carbon source to synthesize N-doped well-organized mesoporous carbon (OMC). OMC had a pore size between 3.5–4.0 nm and a good graphene structure, and these characteristics give it good electrochemical properties. As shown in Figure 4d, in 2018, Chu et al. [66] prepared a cross-coupled large mesoporous carbon material with several excellent properties by using NaNO_3_ salt crystals as a hard template. The prepared material had a highly developed, robust, monolithic 3D interconnected mesoporous framework with an ultra-high SSA close to 2872.2 m^2^ g^−1^, and mesopores size was distributed in 2–4 nm. As shown in Figure 4e, in 2022, Li et al. [67] were able to finely tune the graded porous nanowire materials to obtain the optimal pore structure by adjusting the percentage of SiO_2_ hard template in the electrostatic spinning precursor solution. A reasonable mesoporous structure provided an appropriate SSA (350 m^2^ g^−1^), fast mass transport channel, and rapid electron transport.

**Figure 4 ijms-24-07291-f004:**
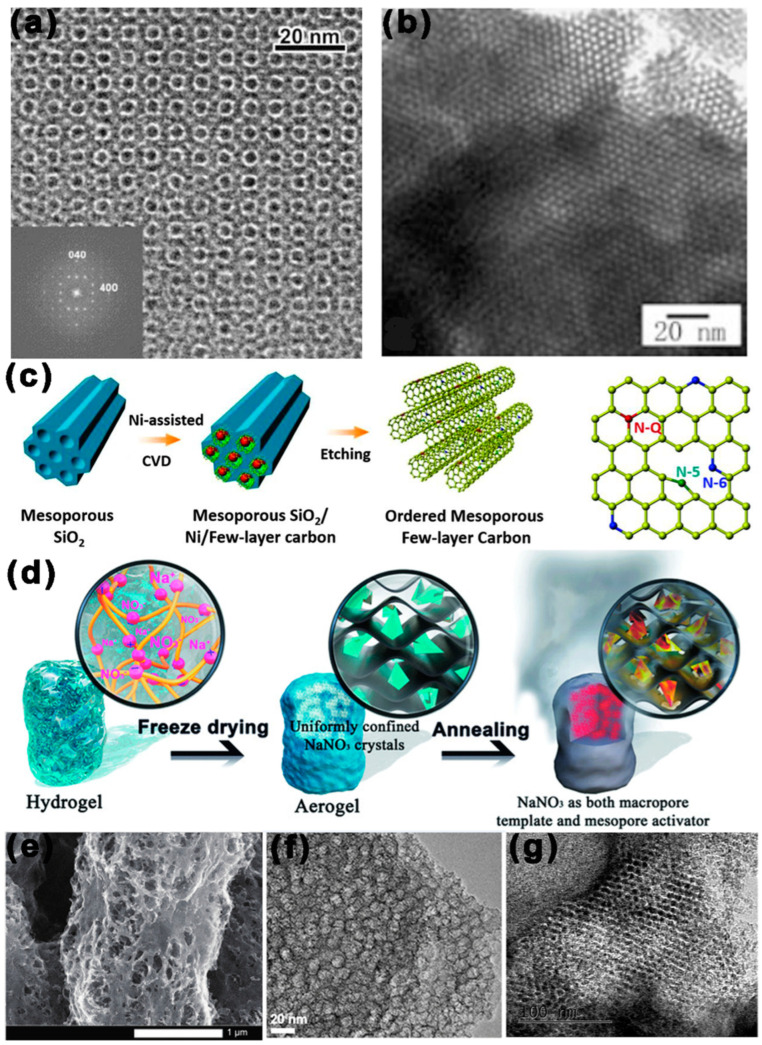
MCBM prepared by the hard templates method. Schematic diagram and TEM images of MCBM based on hard templates: (**a**) TEM image of CMK-2 with SBA-1 as the hard plate [68]. (**b**) TEM image of mesoporous carbon/silica nanocomposites with silica as the hard plate [64]. (**c**) TEM image of ordered mesoporous few-layer carbon with mesoporous SiO_2_ as the hard plate [65]. (**d**) TEM image of a 3D cross-coupled macro–mesoporous carbon network with NaNO_3_ as the hard plate [66]. (**e**) TEM image of porous nanowires-material with SiO_2_ as the hard plate [67]. (**f**) TEM image of CFC@500A with Fe_3_O_4_ nanoparticles as the hard plate [69]. (**g**) TEM image of N, S co-doped mesoporous carbon with SBA-15 as the hard plate [70].

In general, the hard templates method makes the synthesis of MCBM stable and straightforward, especially for those with a high crystalline skeleton. In addition, this method can avoid stringent experimental conditions. The hard templates method also has some shortcomings, such as the precursors struggling to completely fill the mesopores of the hard template because they frequently precipitate outside in the hard template. Therefore, numerous strategies have been investigated to alter the hard template’s surface characteristics, in order to enhance the interaction between precursors and templates.

### 2.3. Soft/Hard Combined Templates

Compared with the hard templates method, the pores generated by the assembly of soft templates generally range from micro to small mesoporous, and it is difficult to achieve the regulation of a large mesoporous scale. In order to gain MCBM with large mesopores to meet the needs of various applications, the soft/hard combined templates method has been developed. AS shown in Figure 5a, in 2007, Zhao et al. [61] adopted PEO–PPO–PEO and monodisperse silica colloidal crystal as a soft and hard template respectively, while resols acted as a carbon precursor. After filling, evaporation-induced self-assembly, and etching, the obtained porous carbon had a face-centered cubic (fcc) structure with abundant mesopores, a large SSA (760 m^2^ g^−1^), and a large TPV (~1.2 cm^3^ g^−1^).

As shown in Figure 5c, in 2009, Aleksandra Zawislak et al. [71] used tetraethyl orthosilicate (TEOS) and colloidal silicon dioxide as hard templates, PEO–PPO–PEO as a soft template, and resorcinol and formaldehyde as carbon sources to synthesize mesoporous carbon. This had a larger SSA of 1730 m^2^ g^−1^, and small mesopores (d = 4 nm) coexist with larger mesopores in the diameter of 11–15 nm. Smaller mesopores arise from the volatilization of the soft template, and larger mesopores arise from the removal of the hard template SiO_2_. In 2015, Chen et al. [72] synthesized nitrogen-modified mesoporous C (N-MCs) using F127 and SiO_2_ nanospheres as soft and hard templates, respectively, while phenolic polymer acted as a carbon precursor. A structural analysis shows that N-MCs have a large SSA (611.9 m^2^ g^−1^) and abundant mesopores size (d = 10 nm). As shown in Figure 5e, in 2018, Liu et al. [73] used F127 and MgAl-layered double hydroxide as soft and hard templates, respectively, while phenolic resin acted as a carbon source to prepare organized mesoporous carbon sheets (OMCSs), which had a large SSA (714 m^2^ g^−1^), a uniform pore size (d = 10 nm), and a larger TPV (0.94 cm^3^ g^−1^).

**Figure 5 ijms-24-07291-f005:**
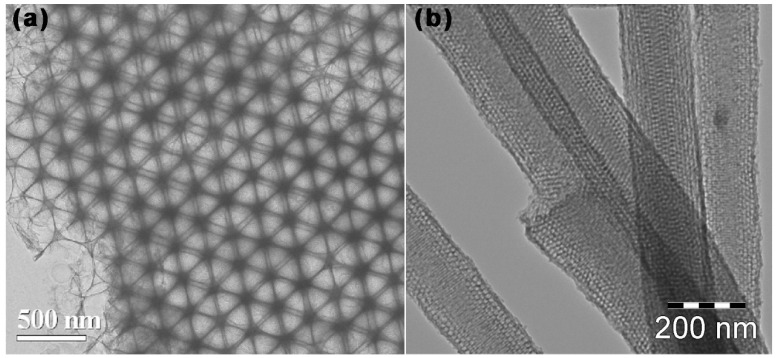
MCBM prepared by the hard templates method. Schematic diagram and TEM images of MCBM based on soft/hard templates: (**a**) TEM image of ordered macro-/mesoporous carbons with Silica colloidal crystals and PEO–PPO–PEO as hard/soft template [61]. (**b**) TEM image of mesoporous carbon nanofibers with AAO membrane and F127 as hard/soft template [74]. (**c**) SiO_2_ and PEO–PPO–PEO for mesoporous carbon [71]. (**d**) SiO_2_ nano-array and P123 for 3D mesoporous carbons [75]. (**e**) LDH and F127 for ordered mesoporous carbon sheets [73].

### 2.4. Template-Free Method

The template-free approach is also frequently used to create mesopores distributed randomly, which are more prevalent than ordered mesoporous carbon structures. In 2009, Dai et al. [76] prepared mesoporous carbon via simple thermolysis of task-specific ionic liquid (TSIL) precursors. By adjusting the cationic/anion pairing properties in TSIL, the porosity and SSA can be controlled precisely. The nitrogen-doped porous carbon porous carbon was rich in mesopores, with a large SSA (780.6 m^2^ g^−1^). As shown in Figure 6b, in 2013, Zhang et al. [77] adopted the improved template-free Pechini method to prepare highly mesoporous carbon foam. Citric acid and Mg(NO_3_)_2_ were used for pyrolysis, and the pyrolysis products were retained as a carbon source. Mesopores were produced by the breakdown of Mg(NO_3_)_2_ to MgO during the carbonization process (800 °C), and additional pores might be produced by H_2_O and CO produced by thermal hydrolysis. MCF had a large SSA (1478.55 m^2^ g^−1^), a mesopore diameter of 5 nm, and a large TPV (2.28 cm^3^ g^−1^).

In 2016, Titirici et al. [78] adopted the template-free method to produce mesoporous carbon, based on the carbonization of zinc and calcium citrate. At 350–500 °C, the carbonates decomposed to produce the corresponding oxides (CaO or ZnO), which were removed by pickling, leaving abundant mesopores. The obtained nitrogen-doped carbon had a high SSA (1350 m^2^ g^−1^), a mesopore diameter of 10nm, and a high TPV (1.2 cm^3^ g^−1^). As shown in Figure 6d, in 2018, Pan et al. [79] adopted the simple template-free method by using polyimide as carbon precursor, to prepare various flower-like layered porous carbons. Modulation of the self-assembly of precursors could be achieved by changing the solvent components, thus changing the nanostructure of mesoporous carbon. The obtained hierarchical porous carbon had a high SSA (941 m^2^ g^−1^), a wide pore size distribution from micropores to mesopores, and a large TPV (0.93 cm^3^ g^−1^). As shown in Figure 6e, in 2021, Liu et al. [80] reported an efficient template-free method for the preparation of N and S co-doped mesoporous hollow C nanospheres (N/S-HMCS). The oligomers in the inner center of the sulfur-bridged covalent triazine frameworks (S-CTF) had a strong propensity to break down and volatilize, then migrate to the outer layer during temperature increase, before etching the carbon layer with the gases produced during pyrolysis, resulting in an abundance of mesopores. N/S-HMC (carbonization at 900 °C) had a large SSA (331 m^2^ g^−1^), abundant mesopores (d = 5 nm), and large TPV (0.69 cm^3^ g^−1^).

Synthesis methods of MCBMs, including the soft templates method, hard templates method, soft/hard templates method, and template-free method are reviewed. The synthesis methods of MCBMs in recent years are reviewed. The soft templates method is flexible and simple, and offers great control over the mesoporous structure, as well as the size and morphology of the synthesized MCBMs. However, some soft templates are remodeled at high temperatures, so calcination may destroy the soft templates, and the assembled structures lead to uncontrollable structures. The assembly mechanism of soft templates has been widely discussed, and several assembly mechanisms have been proposed in recent years. It is important to synthesize MCBMs under the guidance of the mechanism to take advantages of the soft template method. Using a porous substance with a specific structure as a template is known as the hard templates method, which is a precursor assembly and growing process in a restricted space. This method can avoid stringent experimental conditions, such as soft template assembly with precursors, and also allows for the synthesis of MCBMs at high temperature, broadening the backbone structure of mesoporous materials. However, the hard templates method is not suitable for large-scale preparation, due to the large number of templates required, complicated preparation procedures, and time-consuming template removal processes. Therefore, it is of great significance to explore cheap, abundant, and easy-to-etch templates for the wider application of the hard templates method.

Compared with the hard template method, the diameter of the pore prepared by the soft template assembly generally ranges from micropore to small mesopore. It is difficult to achieve the large mesoporous scale, but the existence of large mesopores is crucial for the mass transfer and Li^+^ diffusion in LSBs. In order to synthesize MCBMs to meet the demand of LSBs, the soft/hard combined templates method has been developed. It is very effective for constructing multi-level porous carbon materials, but it also has the disadvantages of both the hard and soft template methods. In addition to these template routes, the template-free method is the direct calcination of carbon precursors. The template-free method is widely used in the synthesis of mesoporous materials with a random distribution of pores, with which it is difficult to achieve precise tuning.

**Figure 6 ijms-24-07291-f006:**
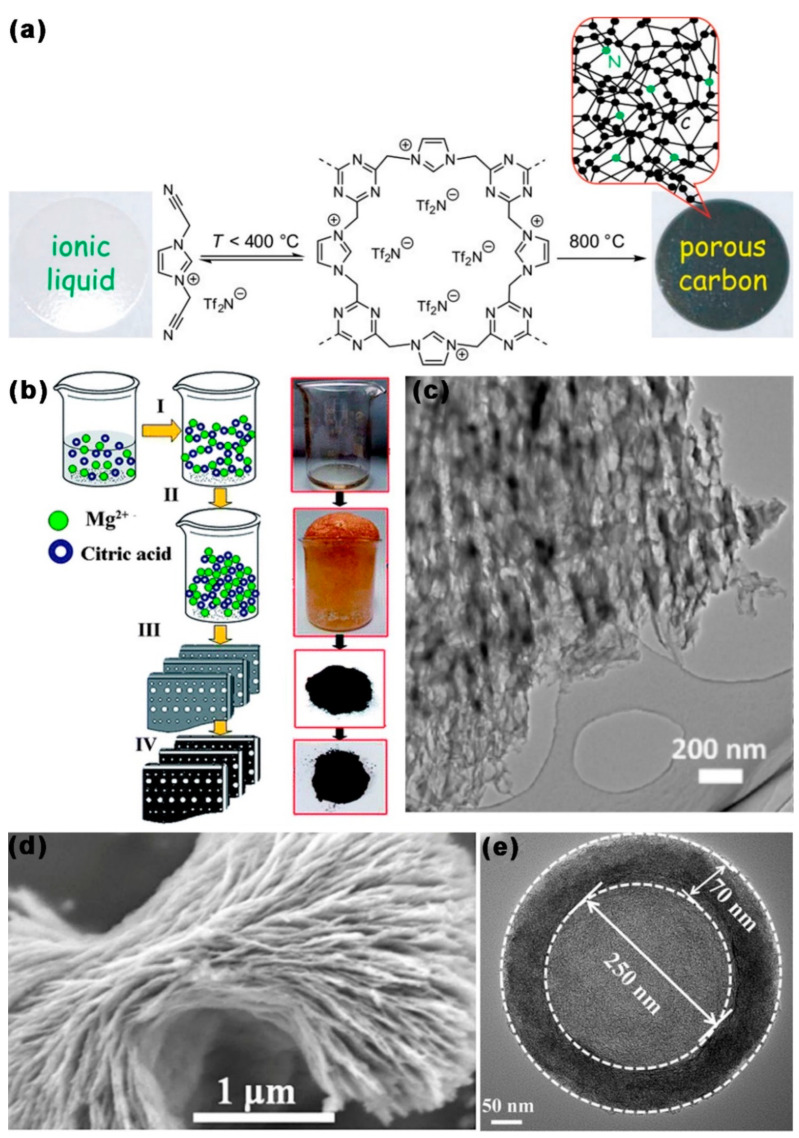
Schematic diagram and TEM images of MCBM prepared by the template-free method: (**a**) The preparation process of [MCNIm][Tf_2_N] [76]. (**b**) The MCF is prepared using the Pechini technique [77]. (**c**) TEM micrographs of NC700 [81]. (**d**) SEM images of FC4 hierarchical porous carbon [79]. (**e**) High-resolution TEM (HRTEM) of the N/S-HMCS900 [80].

## 3. MCBM for Lithium-Sulfur Batteries

### 3.1. Application of MCBM in LSBs Anode

The major barriers to the commercialization of LSBs are low cycle efficiency and concerns about the safety of lithium anodes. Li dendritic redeposition and an uneven distribution of electrolyte disintegration produce on the surface of Li anodes are the primary causes of low cycle efficiency and safety issues. Additionally, Li anodes experience severe corrosion issues as a result of dissolved intermediate LiPSs from the cathode migrating to the Li anode and being converted into insoluble species (like Li_2_S_2_ or Li_2_S), which decreases Li activity and passivates the anode surface [82]. The use of MCBM as a host or modified material to accommodate lithium could greatly slow down the morphological changes of lithium anodes, inhibit the growing of Li dendrites, and enhance the electrical contact between Li and conductor, which is a key research topic with regard to improving the stability of Li anode. The high SSA, strong chemical stability, and low density of MCBM all worked to lower the local current density and extend the life of LSBs.

As shown in Figure 7a, Zheng et al. [83] created a lithiophilic 3D porous collector by adding porous C nanosheets doped with nitrogen (NPCN) with a yolk-shell structure to commercial Li metal. Both the inner and outer surfaces of core-shell NPCN could be used for Li deposition, so NPCN had the capacity to accommodate a large amount of Li. NPCN with a porous structure was conducive to the plating/stripping of Li^+^, thus guiding the uniform lithium nucleation/growth. At 1 mA cm^−2^/1 mAh cm^−2^, the coulomb efficiency (CE) of Li/Cu@NPCN symmetric cells remained 99.0% after 400 cycles, and the charge/discharge curve showed a nucleation overpotential of 18 mV. At 3 mA cm^−2^/1 mAh cm^−2^, the CE of Li/Cu@NPCN symmetric cells remained 98.2% after 250 cycles. The discharge specific capacity (DSC) of the Li/Cu@NPCN|CNT/S full cell was 816 mAh g^−1^, with a CE of 99.9% after 300 cycles at 0.1 C. As shown in Figure 7b, Li et al. [84] used polyacrylonitrile as a carbon source to prepare mesoporous C granules doped with O and N (ONPCGs) with an appropriate SSA (2396 m^2^ g^−1^), a mesoporous diameter of 2.18 nm, and a high TPV (1.3 cm^3^ g^−1^). When ONPCGs were employed as Li’s host material, the high SSA of ONPCGs could effectively reduce the regional electric field and promote stable and uniform Li deposition. At 2 mA cm^−2^/2 mAh cm^−2^, after 350 cycles, the Cu@ONPCG electrode achieved steady Li plating/stripping with a high CE (99%). When up to 20 mA cm^−2^/2 mAh cm^−2^, the CE was 96.4% after 130 cycles. It could cycle steadily for 110 cycles at 30 mA cm^−2^. In addition, the Cu@ONPCG@Li|C/S battery could also reach a DSC of 341 mAh g^−1^ at 2 C and retain a discharge specific capacity of 203.8 mAh g^−1^ after 200 cycles.

As shown in Figure 8a, Lee et al. [85] used mesoporous C (CMK-3) as a Li host, because the large and well-aligned pores of CMK-3 increased the accessibility of Li^+^ and the space for the formation of Li metal, so its electrode had good cycling stability, low polarization, and less Li dendrite formation. At 1 mA cm^−2^/1 mAh cm^−2^, the CE of CMK-3 electrodes symmetric cell was high (>95%) and remained at a high level after 200 cycles. The CMK-3@Li|C/S cell’s initial DSC at 0.5 C was 813.95 mA h g^−1^; after 200 cycles, it was still at 700 mA h g^−1^, with a capacity retention rate of 86% and a capacity degradation rate (CDR) of 0.07% per cycle; the CE of CMK-3@Li was nearly 100% for the first 200 cycles. As shown in Figure 8b, Wan et al. [86] used a 3D hollow C fiber (3D-HCF) container as a Li host, with an SSA of 140.1 m^2^ g^−1^ and a small pore size (0.6–2.5 nm). The high electroactive surface area of 3D-HCF dramatically lowered local current density and improved Li deposition behavior. Li was contained within the hollow fibers and the interstitial area between the fibers. Experimental results showed that there was no unchecked dendrite generation of Li. At 1 mA cm^−2^/1 mAh cm^−2^, the CE of the 3D-HCFs electrode was stable at 99.5% for 350 cycles, with a very low overpotential of 20 mV. At a 1 mA cm^−2^/4 mAh cm^−2^, the CE of the 3D-HCFs electrode stabilized at 99% for 90 cycles. In particular, the CE of the 3D-HCFs electrode-maintained stability at 99% for 75 cycles when increased to 6 mAh cm^−2^.

As shown in Figure 9a, Zhang et al. [87] used a 3D F-doped graphene spindle-shaped embedded MGCN as a host of lithium. First, three-dimensional layered porosity was more suited to volume fluctuations brought on by the stripping and plating of lithium during cycling, assuring the stability of the anode. Second, the high SSA (541 m^2^ g^−1^) of MGCN significantly reduced the local effective current density, thus inhibiting the growing of lithium dendrites in initial nucleation stage. There were abundant mesopores in the carbon branch of MGCN, with a diameter of 3 nm, which was conducive to ion migration. The doping of F was crucial in sustaining solid-electrolyte interphase (SEI), while the graphene structure helped to increase electron conductivity, and had a positive impact on the nucleation and development of Li. At 0.5 mA cm^−2^/1 mAh cm^−2^, the CE of the MGCN electrode was stable at 99% for 300 cycles, with a very low overpotential (<5 mV). At 2 mA cm^−2^/1 mAh cm^−2^, the CE of the MGCN electrode stabilized at 98% for 150 cycles. As shown in Figure 9b, Wang et al. [33] prepared a 3D-printed C framework doped with nitrogen (3DP-NC) and Zn-MOF precursor as a Li host. The 3DP-NC had a large SSA (869 m^2^ g^−1^) and abundant mesopores (d = 3 nm), which could inhibit the growth of Li dendrites and accommodate a large amount of Li metal, stabilize the anode/electrolyte interface, and reduce the local current density. At 1 mA cm^−2^/10 mAh cm^−2^, the CE of the 3DP-NC electrode was stable at 98.1% for 100 cycles. At 2 mA cm^−2^/10 mAh cm^−2^, the CE of 3DP-NC electrode stabilized at 96.5% for 75 cycles.

In addition to MCBM acting as a host of lithium, MCBM can also be used as an interlayer to regulate the deposition and growth of lithium, resulting in a safe and dendrite-free lithium anode. As shown in Figure 10a, Lyu et al. [88] used a top-down approach to prepare a nanowood structured interface as SEI. The nanowood had an SSA of 33.0 m^2^ g^−1^ and abundant mesopores (d = 2.3 nm), and the synthesized SEI had nanochannels between parallel cellulose-based molecules, which could control the deposition of Li. The connected pores on the nanowood and the abundant lithiophilic groups revealed the fast migration of the Li ions. At 1 mA cm^−2^/1 mAh cm^−2^, the CE of Li/Cu cells with nanowood film was stable at 99% for 400 cycles. At 3 mA cm^−2^/1 mAh cm^−2^, the CE of Li-Cu cells with the nanowood film stabilized at 97% for 200 cycles. As shown in Figure 10b, Guo et al. [89] prepared high graphitized wood frame (GWF) by treating pine wood blocks at 2500 °C. GWF had porous tunnel structure and microvilli, with an SSA of 239 m^2^ g^−1^, abundant mesoporores (d~11 nm), and a small TPV (0.027 cm^3^ g^−1^). Due to GWF’s great conductivity, abundant microvillus, and tunnel restraint effect, when it was used as a multifunctional interlayer, the shuttle effect of polysulfide was inhibited. This effectively prevented the corrosion of the Li anode from Sx^2−^ and reduced the production of insoluble substances (Li_2_S_2_/Li_2_S). At the same time, the GWF preserved the three-dimensional transport network of wood, and the microvilli growth on the inner surface of the tunnel provided more deposition sites for LiPSs. With a CE of 91.7%, the initial DSC of LSBs with a GWF interlayer was 1593 mAh g^−1^. The DSC was reduced to 970 mAh g^−1^ after 100 cycles, with a CDR of 0.10% every cycle. The initial DSC of LSBs with GWF interlayers at 1 C was 897.12 mAh g^−1^. The DSC was lowered to 801mAh g^−1^ after 200 cycles, with a CDR of 0.10% every cycle.

### 3.2. Application of MCBM in LSBs Cathode

There are three main problems with the LSBs cathode. First, the low conductivity of S and Li_2_S results in large overpotential, low rate capability, and low S usage rates. Second, the infamous shuttle effect results in active material loss, a low CE, anode corrosion, and a limited cycle life. Third, during the charge/discharge progress, S will result in an 80% volume expansion, resulting in the collapse of host materials, the shedding of S, and rapid capacity attenuation [35,90,91,92]. The use of MCBM with large SSA and great electrical conductivity as sulfur host is an important way to solve these problems, as the high SSA can enhance the interaction between S and C and offer a large number of active sites for LiPSs adsorption. Mesopores can also accommodate large amount of sulfur and slow down volume change. In addition, because of its high electrical conductivity, MCBM can facilitate electron transmission and enhance S consumption. Physical restrictions on polar LiPSs, however, can only be offered by the pore structure, because carbon is non-polar in nature. The use of mesoporous carbon materials doped with heteroatoms (N, P, O, S, etc.)—or compounded with other polar materials with a catalytic effect (transition metals, oxides, hydroxides, sulfides, nitrides, carbides, phosphides, etc.)—is an effective way of solving these problems. It is also a major research topic at present, and could have a positive impact on promoting the commercialization of LSBs.

#### 3.2.1. Heteroatoms-Doped MCBM

Wang et al. [93] used a one-step dual template technique to synthesize hollow C spheres doped with nitrogen (MHCS). The MHCS had interconnected mesopore shells, a high SSA (1875 m^2^ g^−1^), a high TPV (4.75 cm^3^ g^−1^), and abundant mesopores (d~3.6 nm) formed by the removal of surfactant (CTAB). When MHCS was used as the host of S, the carbon shell of the mesopores provided a larger space for the volume expansion of S in the progress of charge and discharge. While the mesopores would reduce the shuttle effect, the large porosity and SSA provide more adsorption sites while accelerating the ion transport, and N, O doping further enhances the chemisorption of LiPSs. At 0.5 C, the DSC of LSBs with the MHCS/S@P cathode was kept at 475 mAh g^−1^ after 1500 cycles, notably, the DSC was still kept at 409 mAh g^−1^ after 3100 cycles, with a medial CDR of 0.023% per cycle. When the sulfur content was up to 4.1 mg cm^−2^, the initial DSC was 780 mAh g^−1^ at 0.5 C; notably, it still provided a DSC of 432 mAh g^−1^ after 1100 cycles, with a low CDR of 0.054%.

As shown in Figure 11a–d, Sun et al. [94] synthesized mesoporous carbon doped with oxygen (BCP) by balsa waste. The SSA of BCP was 3068.40 m^2^ g^−1^ and the TPV was 1.37 cm^3^ g^−1^. When BCP was used as the host of S, the pore structure of BCP could effectively capture LiPSs to inhibit the shuttle effect, which was demonstrated by the intensity of the absorption peak of Li_2_S_6_ in the UV–Vis absorption spectrum (Figure 11e). Meanwhile, mesopores could promote the diffusion of Li^+^ during cycling, especially in the formation stages of S_8_ and Li_2_S, which could significantly improve the lithium storage capacity. At 0.1 C, the DSC of LSBs with BCP/S-6 cathode was kept at 925 mAh g^−1^ after 100 cycles. At 2 C, the DSC was kept at 375 mAh g^−1^ after 1000 cycles. Palmqvist et al. [95] prepared sulfur-functionalized organized mesoporous C (S-OMC), adopting furfuryl alcohol and furfuryl mercaptan as precursors and SiO_2_ as a hard template. The S-OMC had a high SSA (1011.5 m^2^ g^−1^), abundant mesopores (d = 4 nm), and a high TPV (1 cm^3^ g^−1^). When S-OMC was used as the host of S, the covalent S functional groups inside the mesoporous carbon could provide abundant adsorption sites for S, which was the primary cause of LSBs’ enhanced performance. At 0.1 C, the DSC of LSBs with S-OMC-100S-2 cathode was 517 mAh g^−1^.

#### 3.2.2. Transition Metal Composite MCBM

Transition metals have a strong polarity, which can change the polarity of MCBM, increase the chemical adsorption capacity of LiPSs, and block the shuttle effect. At the same time, the redox process of S can be catalyzed by a few transition metals. Therefore, transition metal composite MCBM as a sulfur host is extensively investigated by scientists.

As shown in Figure 12a–h, Lou et al. [48] prepared a nitrogen-doped mesoporous carbon with a uniform distribution of Ni single atoms (Ni-NC(p)) by using an NH_3_Cl-assisted pyrolysis strategy. ZIF-8 would be etched in situ by HCl, the pyrolysis byproduct of NH_3_Cl limited in ZIF-8, to create abundant mesopores in Ni-NC(p). Ni-NC(p) had a thin outer shell and a porous internal network, with an SSA of 428.8 m^2^ g^−1^, a mesopore pore size of 13 nm, and a TPV of 1.1 cm^3^ g^−1^. The mesoporous structure of Ni-NC(p) offered efficient confinement, mass S loading, and good electron and ion transport channels when utilized as the host of S. The catalytic action of Ni single atoms improved the conversion of LiPSs. At 0.5 C, the initial DSC of LSBs with Ni-NC(p)/S cathode was 966.6 mAh g^−1^; notably, the DSC was still kept at 513.9 mAh g^−1^ after 1600 cycles, with a CDR of 0.078% per cycle. At 2 C, the DSC was as high as 706.27 mAh g^−1^. Wang et al. [99] used NaCl as a template to synthesize three-dimensional porous carbon doped with nitrogen and evenly inserted Co atomic clusters (Co/PNC). Co/PNC had a webbed carbon framework with hierarchical pores, with an SSA of 588.0 m^2^ g^−1^ and abundant mesopores (d~5 nm). When Co/PNC served as a host of S, the mesoporous channel provided complete access to the electrochemical interface of the LiPSs, and exhibited a large number of exposed cobalt implantation sites. At the same time, doping of N and Co inhibited the shuttle effect of LiPSs by chemisorption. At 0.5 C, the initial DSC of LSBs with a Co/PNC/S cathode was 1105.4 mAh g^−1^; notably, the DSC was still kept at 746.7 mAh g^−1^ after 100 cycles. At 1 C, the initial DSC was 540.6 mAh g^−1^ and kept at 436 mAh g^−1^ after 100 cycles with a CDR of 0.078% per cycle.

As shown in Figure 12i, by co-doping Ni nanoparticles and N atoms into a network of CNTs, Zheng et al. [100] created a Ni catalyzed carbonization process to create 3D highly conductive C foams (CFs). CFs had an SSA of 156.8 m^2^ g^−1^, and abundant mesopores pores (d = 9.8–14.7 nm). When CFs with high electrical conductivity were utilized as the host for S, the rich porous structure offered adequate room to retain enough S and encouraged interaction between the interface and the electrolyte. In addition, Ni nanoparticles and nitrogen atoms doped on CFs had strong chemisorption ability to LiPSs, which could accelerate the redox reaction of LiPSs. At 135 mA g^−1^, the initial DSC of LSBs with CF950/S cathode was 855.6 mAh g^−1^; notably, the DSC was still kept at 586.5 mAh g^−1^ after 120 cycles with a S loading of 3.71 mg cm^−2^. At 270 mA g^−1^, the initial DSC of LSBs was 667.6 mAh g^−1^; notably, the DSC was still kept at 565.46 mAh g^−1^ after 100 cycles, with a CDR of 0.153% per cycle.

#### 3.2.3. Metal Oxide Composite MCBM

The polar metal–oxygen bonds in metal oxides usually consist of metal cations and oxygen anions, conferring a large number of polar active sites to adsorb LiPSs. Metal oxides have a high electric density, and using metal oxides as the S host material will potentially increase the volume energy density of LSBs. Therefore, oxides and mesoporous carbon combinations have been extensively investigated by scientists.

Lu et al. [101] loaded Al_2_O_3_ uniformly in a nitrogen-rich mesoporous carbon skeleton (NMC-Al_2_O_3_) as an S host. NMC-Al_2_O_3_ had a spherical morphology and an abundant mesopores structure, with a high SSA of 1485 m^2^ g^−1^, abundant mesopores (d = 12 nm), and a TPV of 2.25 cm^3^ g^−1^, which can accommodate more S. Density functional theory (DFT) calculation and experimental results revealed that the chemical adsorption and catalytic effect could be greatly enhanced by doping Al_2_O_3_, which could effectively prevent the shuttle effect, thus significantly enhancing the utilization rate, reversibility, and stability of the S cathode. At 0.5 C, the DSC of LSBs with an NMC-Al_2_O_3_/S cathode was 902 mAh g^−1^; notably, the DSC was still kept at 685 mAh g^−1^ after 1000 cycles, with a CDR of 0.023% per cycle. As shown in Figure 13e, Long et al. [102] used the solgel method to anchor ultra-thin silica nanoparticles uniformly on large mesoporous carbon skeleton (SMC). SMC had a high SSA of 365.45 m^2^ g^−1^, a mesopore pore size of 50 nm, and a TPV of 0.473 cm^3^ g^−1^. Excellent electrical conductivity was provided by the extensive mesopore network of SMC. Silica nanoparticles might significantly increase the polarity of the mesoporous carbon skeleton to increase the loading capacity of S. In addition, the chemical adsorption of silica nanoparticles to LiPSs also significantly inhibited the shuttle effect. It was also demonstrated that SMC/SiO_2_ possessed the strongest adsorption capacity by systematic DFT theoretical calculations (Figure 13e). At 0.2 C, the initial DSC of LSBs with an SMC/S cathode was 969.7 mAh g^−1^; notably, the DSC was still kept at 625.5 mAh g^−1^ after 400 cycles, with a CDR of 0.088% per cycle.

#### 3.2.4. Metal Hydroxides Composite MCBM

Another type of polar compound with numerous hydrophilic/hydroxyl groups are metal hydroxides, which could capture LiPSs effectively. Additionally, the redox dynamics of S can be improved by the electrocatalytic activity of metal hydroxides. Therefore, scientists have conducted extensive research on the composites of metal hydroxides and mesoporous carbon.

As shown in Figure 14a–d, Wang et al. [104] anchored tin hydroxide quantum dots (TOH) on a cellular porous C (HPC@TOH), as the host of S. HPC@TOH had a porous construction, with an SSA of 132.99 m^2^ g^−1^. The results of experiments and DFT theoretical calculations showed that TOH could improve the chemisorption ability, reduce the surface activity of redox reaction, and promote the transformation of LiPSs. Thus, by the synergy effect of physical capture and chemisorption, as well as catalytic conversion, at 0.5 C, the DSC of LSBs with HPC@TOH/S cathode was 1023.74 mAh g^−1^; notably, the DSC was still kept at 773.86 mAh g^−1^ after 200 cycles. At 1 C, the initial DSC of LSBs was 918.05 mAh g^−1^, the DSC was still kept at 697.72 mAh g^−1^ after 400 cycles, with a CDR of 0.06% per cycle.

As shown in Figure 14e–g, Wu et al. [105] uniformly coated a network of Ni_3_(NO_3_)_2_(OH)_4_ on a mesoporous C synthesized by nodus nelumbinis rhizomatis (NNH/PC) as a host for S. NNH/PC had a high SSA of 2615 m^2^ g^−1^ and abundant micro/mesopores (d = 1–50 nm). The reticular NNH layer with good electrical conductivity provided a powerful chemisorption to LiPSs by a nickel–sulfur bond, and it also provided sufficient active sites for the redox reaction of S and inhibited the shuttle effect. Consequently, at 0.5 C, the initial DSC of LSBs with NNH/PC/S cathodes was 1203.5 mAh g^−1^; notably, the DSC was still kept at 521.3 mAh g^−1^ after 700 cycles, with a CDR of 0.081% per cycle.

#### 3.2.5. Metal Sulfides Composite MCBM

Another typical category of polar inorganic compounds are metal sulfides. Metal sulfides have high electrical conductivity, some of them even exhibit metallic or semi-metallic phases. Additionally, metal sulfides have a great affinity for S species, giving LiPSs a powerful chemisorption capacity. Numerous studies have been carried out on metal sulfide as a sulfur host for LSBs.

As shown in Figure 15a–d, Ran et al. [106] prepared a novel cobalt-based double catalytic site co-doped mesoporous carbon (Co_9_S_8_-NSHPC). The SSA of Co_9_S_8_-NSHPC was 521.42 m^2^ g^−1^, with abundant mesopores (d = 2–30 nm). Experimental and DFT calculations results showed that the designed Co9S8-NSHPC had moderate adsorption energy for LiPSs, which was favorable for the reversible catalytic conversion. In addition, DSCSs enhanced the catalytic kinetics and facilitated the redox reaction of S. Consequently, at 0.2 C, the initial DSC of LSBs with Co9S8-NSHPC cathode was 918 mAh g^−1^; notably, the DSC was still kept at 867 mAh g^−1^ after 200 cycles, with a CDR of 0.028% per cycle. As shown in Figure 15e, Zhang et al. [107] fixed MoS_2_ nanosheets containing sulfur vacancies in hollow mesoporous C (MoS_2−X_/HMC) through a S–C bond. The SSA of MoS_2−X_/HMC was 146.6 m^2^ g^−1^ and the TPV was 1.31 cm^3^ g^−1^. Ultra-thin MoS_2_ sheets with a mass of S vacancy had strong chemisorption for LiPSs, which could catalyze the rapid redox conversion of LiPSs and enhance the reaction kinetics. Consequently, at 0.2 C, the initial DSC of LSBs with MoS_2−X_/HMC/S cathode was 1077 mAh g^−1^; notably, the DSC was still kept at 754 mAh g^−1^ after 100 cycles, with a CDR of 0.3% per cycle. When up to 1 C, the initial DSC of LSBs with MoS_2−X_/HMC/S cathode was 920 mAh g^−1^; notably, the DSC was still kept at 530 mAh g^−1^ after 500 cycles, with a CDR of 0.085% per cycle.

#### 3.2.6. Metal Nitrides Composite MCBM

Metal nitrides are another polar substance with excellent electrical conductivity. Metal nitrides and mesoporous C composites are still in the earliest stages of development for LSBs, because of their complex synthesis process.

As shown in Figure 16a–c, Chen et al. [110] loaded titanium nitride (TiN) nanoparticles uniformly in mesoporous C doped with nitrogen (RF-TiN). The SSA of RF-TiN was 900 m^2^ g^−1^, the mesopores size was 42 nm, and the TPV was 4.12 cm^3^ g^−1^. RF provided abundant physical adsorption sites for LiPSs. RF also restricted TiN with stronger polarity and higher conductivity (5 × 10^6^ S m^−1^) to the nanoscale, preventing the agglomeration of TiN nanoparticles. In addition, RF completely exposed the TiN adsorption surface, and the adsorption sites were highly dispersed, which inhibited the shuttle effect of LiPSs. Hence, at 1 C, the initial DSC of LSBs with RF-TiN/S cathode was 924 mAh g^−1^; notably, the DSC was still kept at 700 mAh g^−1^ after 800 cycles, with a CDR of 0.04% per cycle. As shown in Figure 16d–i, Sun et al. [111] embedded VN quantum dots uniformly in the mesoporous hollow carbon spheres (VN-H-C), which had an SSA of 316.42 m^2^ g^−1^ with abundant mesopores (d = 5–20 nm). LiPSs could be physically and chemically adsorbed onto the VN-H-C with excellent efficiency, and the conversion of LiPSs could be greatly speed up by VN. Hence, at 1 C, the initial DSC of LSBs with VN-H-C/S cathode was 856.5 mAh g^−1^; notably, the DSC was still kept at 602.5 mAh g^−1^ after 500 cycles, with a CDR of 0.059% per cycle.

#### 3.2.7. Metal Carbides Composite MCBM

Metal carbides are another polar conducting material with extremely high potential for cathode applications in LSBs. Research on metal carbides as S hosts for LSBs, however, is still in its early stages, because of the harsh conditions needed to produce metal carbide-MCBM.

Using a Trichoderma bioreactor and an annealing procedure, As shown in Figure 17a–e, Wu et al. [113] created an N, P, Co-doped trichoderma spore carbon (TSC) structure implanted with NbC nanoparticles (TSC/NbC). TSC/NbC with a bowl shape had a large SSA (555 m^2^ g^−1^) and abundant mesopores (d~30–50 nm). S could be easily tolerated in TSC/NbC due to NbC’s twin responsibilities in TSC, which included creating pores and improving conductivity. The porous conductive structure enhanced the physical adsorption of LiPSs, and the doping of N, P elements and polar conductive NbC in TSC enhanced the chemical adsorption of polysulfide. So, at 0.1 C, the initial DSC of LSBs with TSC/NbC/S cathode was 1153.63 mAh g^−1^; notably, the DSC was still kept at 937.9 mAh g^−1^ after 500 cycles, with a CDR of 0.037% per cycle. Yan et al. [114] created a novel S host by combining hollow C spheres with abundant mesoporous channels with a few very thin Mo_2_C/C nanosheets (MHCS@Mo_2_C/C). The SSA of MHCS@Mo_2_C/C was 813.6 m^2^ g^−1^, the pore diameter distribution was 3–9 nm, and the TPV was 1.17 cm^3^ g^−1^. Mo_2_C/C nanosheets might decrease the electron/ion transport pathway and improve LiPSs’ chemisorption. MHCS sped up electron/ion movement, improved the physical adsorption of LiPSs, and had a buffering effect on the volume expansion of S. At 0.1 C, the initial DSC of LSBs with a TSC/NbC/S cathode was 1316.8 mAh g^−1^; notably, the DSC was still kept at 880.1 mAh g^−1^ after 100 cycles. At 1 C, the initial discharge DSC was kept at 874.1 mAh g^−1^ after 500 cycles, with a medial CDR of 0.045% per cycle.

#### 3.2.8. Metal Phosphates Composite MCBM

Metal phosphides are polar materials with the highest electrical conductivity relative to its oxides and sulfides. In addition, the synthesis conditions for metal phosphides are simpler and milder than nitrides and carbides. Metal phosphates composite MCBM have gained a lot of attention in recent years.

As shown in Figure 18a–d, by phosphating and pyrolyzing acrylic resin, Shen et al. [116] created a mesoporous 3DC framework that was implanted with CoP nanoparticles (CoP@3DC). The SSA of CoP@3DC was 867.78 m^2^ g^−1^, the pore diameter distribution was 5–15 nm. The mesoporous construction of CoP@3DSC inhibited the shuttle effect, and the high conductivity of 3DC was conducive to the electron transport. The uniformly distributed CoP nanoparticles in 3DC improved the chemisorption and conversion of LiPSs to Li_2_S. Consequently, at 0.5 C, the initial DSC of LSBs with CoP@3DC/S cathode was 1117.37 mAh g^−1^; notably, the DSC was still kept at 740.56 mAh g^−1^ after 600 cycles, with a CDR of 0.056% per cycle. At 1 C, the initial DSC was 936.80 mAh g^−1^, and the DSC was still kept at 640.00 mAh g^−1^ after 500 cycles, with a CDR of 0.0528% per cycle. As shown in Figure 18e–i, Hu et al. [117] embedded FeP nanoparticles into hierarchical porous C microspheres (PCM/FeP). The SSA of PCM/FeP was 500 m^2^ g^−1^, the mesoporous diameter was 4 nm, and the TPV was 0.9 cm^3^ g^−1^. Under the restriction of mesopores, PCM might strengthen the interaction between FeP and S species and increase the catalytic sites exposure of FeP nanoparticles. FeP could improve the chemisorption of LiPSs and optimize the transformation kinetics of LiPSs to Li_2_S. Consequently, at 0.5 C, the initial DSC of LSBs with a PCM/FeP/S cathode was 1231 mAh g^−1^; notably, the DSC was still kept at 910.7 mAh g^−1^ after 500 cycles, with a CDR of 0.05% per cycle.

As shown in Table 1, extensive study on MCBMs as LSBs cathode hosts has been conducted, and considerable advancements have been made. Large SSA and excellent electrical conductivity of MCBMs are key components in enhancing S and Li2S’s poor conductivity and accommodating the volume expansion. By combining heteroatoms, transition metals, oxides, hydroxides, sulfides, nitrides, carbides, and phosphides, MCBMs have progressed from their original roles as physical adsorption and spatial domain limiting to a range of high-polar, high-conductive, and effective catalysts for S redox processes. 

### 3.3. Application of MCBM in Separators or Interlayers

The fundamental elements of LSBs that are crucial to battery performance are the separators, which act as an insulator and ion transport medium between Li anode and S cathode. One of the most efficient ways to reduce the shuttle effect, speed up the S redox reaction, and enhance the performance of LSBs is to incorporate an MCBM interlayer between the S cathode and separator, or to construct a separator optimized by MCBM.

#### 3.3.1. MCBM-Coated Separators for LSBs

As shown in Figure 19a–c, Park et al. [145] anchored NiS_2_-MnS on a N-rich 3D linked hollow graphene-C spheres with homogeneously distributed MoS_2_ (NiS_2_-MnS/MoS_2_-3DNGr). The product had an SSA of 420.9 m^2^ g^−1^ and abundant mesopores (d = 3.4–9.65 nm). MoS_2_ and NiS_2_-MnS had good adsorption capacity and catalytic conversion ability for LiPSs. In addition, 3DNGr ensured the uniform dispersion of NiS_2_-MnS nanoparticles and MoS_2_ nanosheets, which accelerated the transport of electrons to the catalytic active site. Concurrently, the mesoporous structure of 3DNGr facilitated the diffusion of lithium ions. Thus, at 0.1 C, the initial DSC of LSBs with a NiS_2_-MnS/MoS_2_-3DNGr-optimized separator was 1011 mAh g^−1^; notably, the DSC was still maintained at 768.36 mAh g^−1^ after 200 cycles, with a CDR of 0.12% per cycle. At 3 C, the DSC was still kept at 798 mAh g^−1^, with a CDR of 0.035% per cycle. As shown in Figure 19d–k, an in situ catalysis and a simple self-templating procedure were used by Xiong et al. [146] to synthesize mesoporous C co-doped with Co and Ni (CoNi@MPC), which were directly coated on a separator. The SSA of CoNi@MPC was 190.6 m^2^ g^−1^, the mesopores diameter was 4.8 nm, and the TPV was 0.23 cm^3^ g^−1^. MPC as a conductive skeleton was conducive to the electronic conductivity. Meanwhile, the Co-Ni active sites uniformly distributed on MPC could catalyze the transformation of LiPSs, an in particular accelerate the formation and decomposition of solid product of Li_2_S, effectively alleviating the shuttle effect. At 1 C, the initial DSC of LSBs with a CoNi@MPC-coated separator was 1050.82 mAh g^−1^; notably, the DSC was still maintained at 724.7 mAh g^−1^ after 500 cycles, with a CDR of 0.090% per cycle. At 4 C, the DSC was still kept at 580.7 mAh g^−1^ after 500 cycles, with a CDR of 0.087% per cycle. Liu et al. [147] synthesized a mesoporous C/TiO_2_ (MC/TO) composite material as the surface modification material of separator. MC/TO contained abundant mesopores with a SSA of 81.2 m^2^ g^−1^. Polar metal oxide TiO_2_ on MC/TO coated separator enhanced the chemical adsorption of LiPSs, while MC could improve the electrical conductivity. Thus, at 2 C, the initial DSC of LSBs with an MC/TO-coated separator was 816 mAh g^−1^; notably, the DSC was still kept at 658 mAh g^−1^ after 500 cycles, with a CDR of 0.12% per cycle.

#### 3.3.2. MCBM Interlayers for LSBs

As shown in Figure 20a–k, By using a straightforward coprecipitate technique, Peng et al. [148] implanted CoP nanoparticles into porous C spheres (CoP@C). CoP@C had an SAA of 128.4 m^2^ g^−1^, a large number of active sites to improve LiPSs chemisorption, and CoP had strong electrocatalytic activity, which could reduce the conversion energy barrier and, ultimately, promote the kinetics of LiPSs transformation. Consequently, at 0.2 C, the initial DSC of LSBs with CoP@C interlayer was 1213 mAh g^−1^; notably, the DSC was still kept at 1015 mAh g^−1^ after 100 cycles. At 2 C, the DSC was still kept at 550 mAh g^−1^ after 800 cycles, with a CDR of 0.058% per cycle. As shown in Figure 20l,m, Guo et al. [89] used a high-graphitized wood frame (GWF) as the LSBs’ interlayer. GWF had a porous tunnel structure and a large number of microvilli, with a small SSA of 2.4 m^2^ g^−1^. The pore size distribution was concentrated in the range of 20–30 nm, and it had a small TPV of 0.0057 cm^3^ g^−1^. GWF had a high conductivity and optimized 3D transport network. Microvilli growing on the inner surface of GWF tunnels not only provided more deposition sites for LiPSs, but also effectively inhibited the shuttle effect through the same principle as nose hair filtering air. At the same time, the GWF interlayer could protect the lithium anode from the corrosion of Sx^2−^ effectively, so the lithium anode could maintain structural stability during the long cycle. Consequently, at 1 C, the initial DSC of LSBs with GWF interlayer was 801 mAh g^−1^; notably, the DSC was still kept at 705 mAh g^−1^ after 200 cycles, with a CDR of 0.06% per cycle. Kang et al. [149] synthesized a flexible activated C nanofibers (ACNF) with an adjustable pore structure by combining electrospinning polyimide with activation treatment. ACNF had a porous structure with an SSA of 995 m^2^ g^−1^, abundant mesopores (d = 2.5–10 nm), and a TPV of 0.51 cm^3^ g^−1^. When the ACNF acts as the interlayer of LSBs, the highly conductive ACNF effectively intercepted the shuttle effect, and it also accelerated the ion transport. Consequently, at 0.1 C, the initial DSC of LSBs with ACNF interlayer was 1224 mAh g^−1^; notably, the DSC was still kept at 897 mAh g^−1^ after 100 cycles.

### 3.4. Application of MCBM as Two-in-One Hosts

The above-mentioned notorious shuttle effect, slow reaction kinetics in the S cathode, and the growth of Li dendrites in the Li anode are major roadblocks to the commercialization of LSBs. As a result, tackling the issues with cathodes and anodes at the same time is an effective way to create practical LSBs. However, reports of hosts that could serve as both anode and cathode were infrequent. In this context, scientists have developed functional host materials that serve as both Li anodes and S cathodes, defining such materials as two-in-one carriers. Compared with other conventional electrodes, electrodes using MCBM as a two-in-one carrier have the following advantages: (1) a high porosity structure and TPV, which can accommodate Li and S and mitigate the volume variation of electrodes caused by the volume expansion of S and Li dendrites; (2) large SSA, which slows lithium dendrite formation and lowers local current density during Li stripping/plating, and keeps the anode in a dendrite free state; (3) strong physical/chemical adsorption to LiPSs, which suppresses the shuttle effect to a certain extent; (4) the optimized mesoporous channels speed up the transportation of Li^+^ and electrons, thus improving the utilization rate of S.

As shown in Figure 21a–c, Zhu et al. [150] designed a bi-functional tunable mesoporous carbon sphere (MCS) that served as a two-in-one host. The prepared MCS had a high SSA of 1340 m^2^ g^−1^, and was abundantly mesoporous (d = 2.8–20 nm). Due to the honeycomb structure of MCS and its abundance at nitrogen sites, Li^+^ was uniformly deposited, which prevented the growth of Li dendrites. At 1 mA cm^−2^/1 mAh cm^−2^, the CE of Li@MCS900 electrode remained 96.0% after 300 cycles. MCS was able to pack S uniformly inside and reduced the shuttle effect. At 2 C, the initial DSC of LSBs with S@MCS900 cathode was 411 mAh g^−1^, and the DSC was still kept at 400 mAh g^−1^ after 200 cycles. LSBs with S@ MCS900 as a cathode and Li@MCS900 as an anode could be cycled stably for 200 cycles with a high DSC retention rate at 1 C. As shown in Figure 21d–h, Wu et al. [151] synthesized hierarchical Co and N co-doped mesoporous C nanosheets (Co/N-PCNSs) as two-in-one hosts for Li and S. Co/N-PCNSs had a high SSA of 500 m^2^ g^−1^, a medial mesopore diameter of 38 nm, and a TPV of 1.03 cm^3^ g^−1^. It was possible to successfully regulate the Li plating and stop the growth of Li dendrites thanks to the rich mesoporous structure and high lithophilicity increased by Co and N. At 0.5 mA cm^−2^/0.5 mAh cm^−2^, the Li|Li@Co/N-PCNSs symmetric batteries were able to cycle stably for 350 h. The encapsulated Co nanoparticles could synergize with nitrogen heteroatoms to absorb LiPSs and promote the conversion of S species in cathodes. At 5 C, the DSC of LSBs with S@Co/N-PCNSs cathode was kept at 411 mAh g^−1^ after 400 cycles, with a CDR of 0.023% per cycle. LSBs with a S@Co/N-PCNSs cathode and a Li@Co/N-PCNSs anode could be stably cycled for 60 cycles with a high DSC retention rate at 0.2 C. As shown in Figure 21i, Liu et al. [152] distributed Ni nanoparticles uniformly on the O-doped mesoporous C nanofiber framework (Ni@CNF-O) as the two-in-one hosts of Li and S. Ni@CNF-O had an SSA of 153.3 m^2^ g^−1^, the mesopore size was located between 4 and 10 nm, and it had a TPV of 0.288 cm^3^ g^−1^. The regulating electric field generated by the oxidized C mesoporous structure could effectively promote the homogeneous deposition and growth of Li at the nanofibers and finally the entire electrode level. At 0.5 mA cm^−2^/1 mAh cm^−2^, the Ni@PCNF-O@Li symmetric battery could cycle stably for 1200 h. At the same time, the chemisorption of LiPSs by oxygen-containing groups greatly alleviated the shuttle effect and improved the cycle life. At 0.2 C, the initial DSC of LSBs with a Ni@PCNF-O@Li_2_S_6_ cathode was 1206 mAh g^−1^; notably, the SC was still kept at 1080 mAh g^−1^ after 100 cycles. At 1 C, the initial DSC of LSBs with Ni@PCNF-O@Li_2_S_6_ cathode and Ni@PCNF-O@Li anode was 1011 mAh g^−1^, and the DSC was still kept at 996 mAh g^−1^ after 300 cycles, with a CDR of 0.015% per cycle.

As shown in Figure 22a–d, Wang et al. [153] embedded ZnSe-CoSe_2_ in the yolk-shell nitrogen doped carbon framework (NC) as two-in-one hosts. The SSA of ZnSe-CoSe_2_@NC was 94.45 m^2^ g^−1^, the mesopore size was located from 22 to 34 nm, and the TPV was 0.34 cm^3^ g^−1^. Co and Zn distributed in a C framework could guide the homogeneous nucleation of Li, effectively inhibiting the growing of lithium dendrites. At 1 mA cm^−2^/1 mAh cm^−2^, the Li/ZnSe-CoSe_2_@NC symmetric batteries could cycle stably for 1000 h. The ZnSe-CoSe_2_ heterostructure also exhibited high chemisorption, outstanding catalytic effect for LiPSs, and excellent conductivity. At 1 C, the DSC of LSBs with S/ZnSe-CoSe_2_@NC cathode was kept at 619.4 mAh g^−1^ after 400 cycles. At 1 C, the initial DSC of LSBs (S/ZnSe-CoSe_2_@NC||Li/ZnSe-CoSe_2_@NC) was 909.7 mAh g^−1^; notably, the DSC was still kept at 731.8 mAh g^−1^ after 400 cycles, with a CDR of 0.049% per cycle. At 2 C, the DSC was still maintained at 477 mAh g^−1^ after 1000 cycles.

As shown in Figure 22e,f, Li et al. [154] dispersed Ni_2_P nanoparticles uniformly on hollow C spheres (Ni_2_P-HCS) as a Li host and separator modifier. The SSA of Ni_2_P-HCS was 45.7 m^2^ g^−1^, and the mesopores diameter was 14 nm. Ni_2_P-HCS had a lipophilic surface and high ionic/electronic conductivity, which could regulate Li deposition behavior. At 1 mA cm^−2^/1 mAh cm^−2^, the Ni_2_P-HCS electrode was able to cycle stably for 100 cycles with a stable CE over 99%. Ni_2_P-HCS had a polar surface and a strong electrical conductivity, which led to the strong adsorption and catalytic effect. At 1 C, the initial DSC of LSBs with Ni_2_P-HCS-coated separator was 959.26 mAh g^−1^; notably, the DSC was still kept at 781.8 mAh g^−1^ after 500 cycles, with a CDR of 0.037% per cycle. At 1 C, the DSC of LSBs (KB@S||Ni_2_P-HCS@PP||Ni_2_P-HCS@Li) was kept at 700.4 mAh g^−1^ after 300 cycles, with a CDR of 0.015% per cycle. When the S loading was up to 4.6 mg cm^−2^, the DSC was 664.3 mAh g^−1^ at 2 C.

As shown in Figure 22g,h, Wong et al. [155] uniformly loaded BN nanosheets onto C nanotubes (f-BNNSs/f-CNTs) as two-in-one hosts of Li and S. The product had a layered cellular porous structure, with an SSA of 154.9 m^2^ g^−1^. The growth of Li dendrites was inhibited by the uniform electric field generated by the three-dimensional porous structure with great thermal conductivity. At 1 mA cm^−2^/1 mAh cm^−2^, the Li@f-BNNSs/f-CNTs electrode was able to cycle stably for 1000 h, with a low overpotential of 22 mV. The large SSA of the mesopore structure combines polar BN nanosheets with strong adsorption for LiPSs, which greatly reduced the shuttle effect of LiPSs. At 0.5 C, the DSC of LSBs with Li@f-BNNSs/f-CNTs anode and S@f-BNNSs/f-CNTs cathode was kept at 790.5 mAh g^−1^ after 300 cycles, with a CE close to 100%.

## 4. Scientific Challenges and Future Prospects

Many countries promote the development of new energy batteries through various policies, including subsidies, rebates, and tax exemptions. In some countries, electronic devices and vehicles, etc., purchased by government departments are centered on new energy batteries, and there are strong subsidies for individuals to purchase new energy vehicles. The multi-state Zero Emission Vehicle (ZEV) program is the most important new energy battery policy in the United States. The plan calls for zero-emission vehicles to account for about 8% of vehicle sales in zero-emission states by 2025, and this policy is a huge boost to the development of new energy batteries. Additionally, state subsidy policies are changing with the development of new energy batteries. For example, in China, electric vehicles with a range of more than 150 km are eligible for subsidies, and a range of 400 km is required to receive the highest subsidies. Requirements related to battery performance have also become more stringent: the energy density of the battery must exceed 105 Wh kg^−1^ to receive subsidies, and at least 140 Wh kg^−1^ to receive full subsidies, and subsidies will be gradually phased out [156]. Therefore, the support offered by these favorable policies presents an opportunity capitalize on these current trends and urgently develop safe and practical LSBs. Even while laboratory science has made many important advancements, the creation of LSBs for real-world applications has not been straightforward. Future research should carefully take into account the following industry aims in order to hasten the commercialization of LSBs:Ratio of S in Cathode

An optimized host with high electrical conductivity, large S loading capacity, and catalytic activity must be created in order to lower the amount of inactive fraction in cathodes. The industrial target S content should be greater than 70 wt%.

2.Areal S Loading

The preparation of high S loaded electrodes has a large impact on the utilization rate of S, LSBs performance, and cycling life. Laboratory conditions usually adopt relatively low area S loadings, and the industrial indicator of area sulfur loadings should be larger than 7 mg cm^−2^.

3.Electrolyte dosage

Electrolyte dosage is an important indicator of LSBs. The E/S under laboratory conditions is usually greater than 15 μL mg^−1^. When the sulfur content is greater than 80% and the area sulfur loading is greater than 7 mg cm^−2^, the industrial index of E/S should be kept below 5 µL mg^−1^.

4.Li-S Pouch Cells

Coin and pouch cells are assembled using different techniques, and the coin cells with good operational capability does not mean that pouch batteries will perform excellently. Therefore, it is of great significance to assemble coin cells into pouch batteries and evaluate their electrochemical performance.

5.Safety

The growth of Li dendrites introduces safety problems, and the optimization of Li anodes using MCBMs to enhance stability is an important direction.

6.Cost

In order to commercialize LSBs, the raw materials used to synthesize MCBMs should be inexpensive and the synthesis process should be simple and stable.

7.All Solid State LSBs

The advantages of low degree Li^+^ dissolution, rare shuttle effect and Li dendritic growing are shared by all solid state LSBs. Since studies of MCBM for all solid state LSBs are still in their early stages, researchers should pay attention to the following stage of research.

## 5. Conclusions

The first prototype LSBs were put forward in the 1960s, and after more than sixty years of research and exploration, especially in the past decade, great progress has been made in studying them [157]. The synthesis progress of high performance MCBMs in recent years, as well as the design and application of MCBMs in the Li anode, S cathode, separator, interlayer, and two-in-one hosts have been reviewed here. 

Compared to other porous materials, mesoporous materials can be modified flexibly for different systems, and the modification methods are relatively mature. Due to the wide application prospect of MCBMs in LSBs, in order to achieve high SSA, high TPV, appropriate pore size and other requirements, the four synthesis methods described in Chapter 2 can be flexibly combined according to the actual requirements.

In-depth research has been carried out on MCBMs as LSBs cathode hosts, as shown in Table 1, and significant progress has been made. The large SSA and great electrical conductivity of MCBMs are important factors for improving the poor conductivity of S and Li_2_S and accommodating the volume expansion of S. MCBMs have evolved from the initial role of physical adsorption and spatial domain limitation to a variety of high-polar, high-conductive, and good catalysts for S redox reactions by compositing heteroatoms, transition metals, oxides, hydroxides, sulfides, nitrides, carbides, phosphides. Combining physisorption, chemisorption, and catalytic conversion is the best way to solve the problem of S cathode side. MCBMs, as the host material of Li, can slow down the volume change of anode, inhibit Li dendrite growth, and provide enough electrical contact between Li and conductor, which is an important area for research when it comes to enhancing the stability of the Li anode. However, most studies on MCBMs used in anodes mainly focus on Li metal batteries, and study of the LSBs’ anode is still in the early stages. Researchers can learn from the study of the Li metal anode to promote the rapid exploitation of a dendrite-free LSBs anode. The separator acts as an insulator and ion transmission medium between the anode and cathode. Inserting MCBMs as an interlayer between the cathode and separator, or designing a separator optimized with MCBMs, as described in Section 3.3, is of great significance for inhibiting the shuttle effect, accelerating ion and electron transmission, accelerating sulfur redox reaction, and finally improving battery performance.

The major challenges facing the commercialization of LSBs are the severe shuttle effect, slow reaction kinetics in the S cathode, and the growth of Li dendrites in the Li metal anode. MCBMs as two-in-one hosts have been developed, which has an excellent ability to adsorb and catalyze S species as the host of S. At the same time, it is a lithiophilic host of Li, which can effectively promote the homogeneous nucleation of Li, inhibiting the generation of Li dendrites. Bifunctional MCBMs that achieve the protection of both cathode and anode simultaneously are the direction that scientists should focus on. Attention should also be paid to the application of theoretical guidance strategies (e.g., DFT theoretical calculation and machine learning), which have great theoretical guidance significance for accelerating the screening progress of MCBMs with high Li^+^ mobility and a catalytic effect of high S redox reaction.

LSBs are currently only used in a select few domains, and there are still several issues to be resolved before they are suitable for more widespread use. This study focuses more on the outcomes of scientific research into the design and development of materials at the laboratory level than it does on advancements in industrial indicators, such as the design of battery poles and battery heat dissipation systems, etc. We believe LSBs will become more prevalent in the energy storage industry because of the tireless work of researchers and close collaboration with commercial companies.

## Figures and Tables

**Figure 1 ijms-24-07291-f001:**
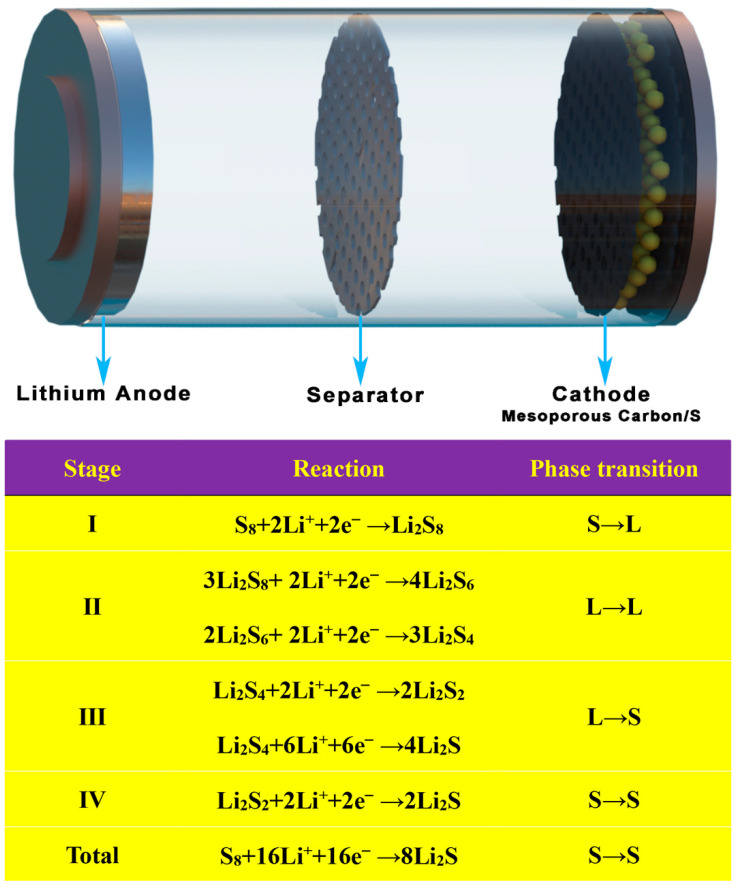
Structure and reaction mechanism diagram of LSBs.

**Figure 2 ijms-24-07291-f002:**
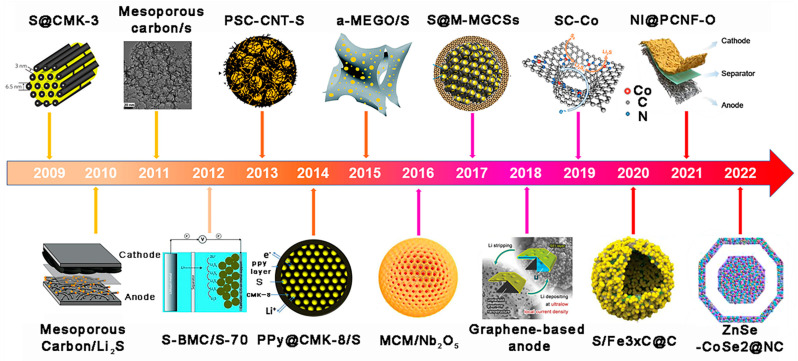
An overview of the history of MCBM and several representative methods for enhancing LSB performance. The representative works inserted in chronological order: S was kept within CMK-3. Over the past ten years, LSBs had had a revival of interest as a result of this effort [35]. Li_2_S was encased within MCBM on the cathode of a Li_2_S/Si battery [36]. Mesoporous carbon with adjustable pore diameters [37]. Bulk bimodal mesoporous carbon (BMC-1) as Li-S cathode [38]. Porous spherical carbon–carbon nanotubes–sulfur composite [39]. The surface of the CMK-8/S was coated with PPy [40]. S deposition on the surface of micro/mesoporous activated graphene (a-MEGO) [41]. Mesoporous carbon microspheres-Nb_2_O_5_ nanocrystals hybrid sulfur cathode system [42]. Sulfur-micro-mesoporous graphitic carbon spheres composite [43]. Li deposited on graphene-based anode [44]. Atomic-cobalt-implanted supramolecule-derived carbon [45]. S was deposited on Fe_3−X_C@C as the S cathode [46]. The oxygen functionalized carbon nanofiber framework with Ni nanoparticles [47]. Carbon frameworks doped with nitrogen in the eggshell that were embedded with ZnSe-CoSe_2_ [48].

**Figure 3 ijms-24-07291-f003:**
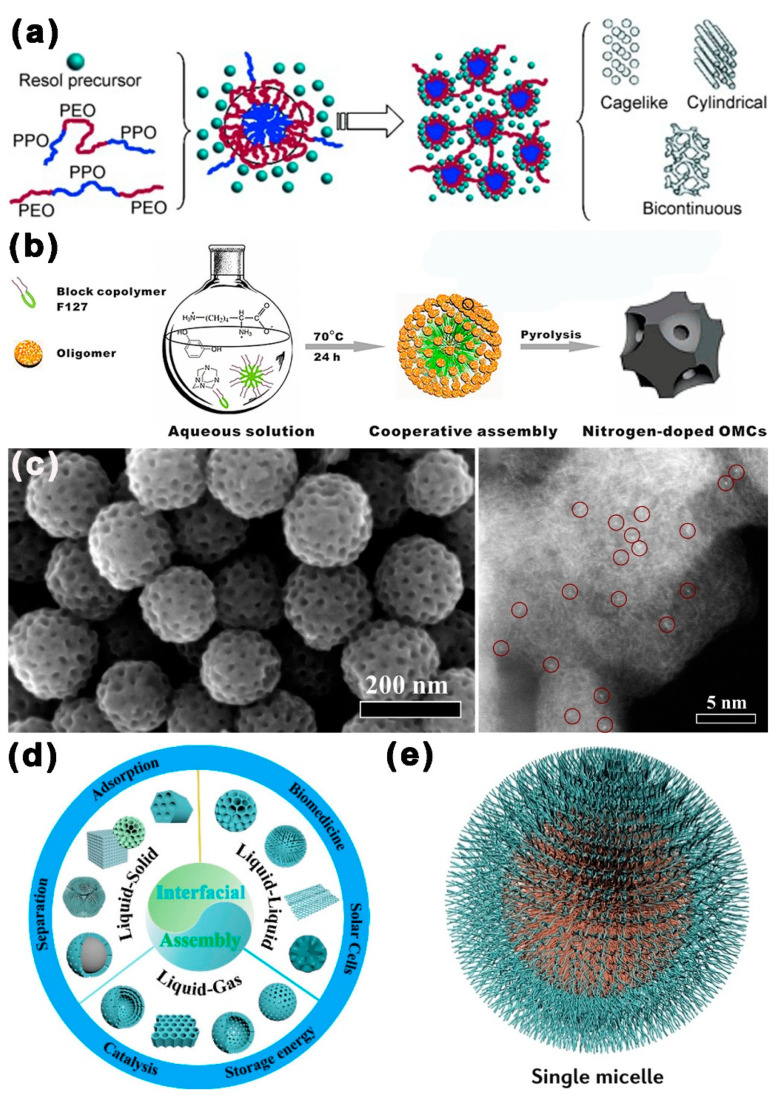
MCBM prepared by the soft template method. Schematic diagram and transmission electron microscopy (TEM) images of MCBM based on soft templates: (**a**) PPO–PEO–PPO for mesoporous carbon sample [52]. (**b**) F127 and P123 for nitrogen-doped OMCs [53]. (**c**) TEM images of meso-Fe-N-C for meso-Fe-N-C [55]. (**d**) Interfacial assembly mechanisms [56] and (**e**) single-micelle-directed synthesis method of mesoporous materials [57].

**Figure 7 ijms-24-07291-f007:**
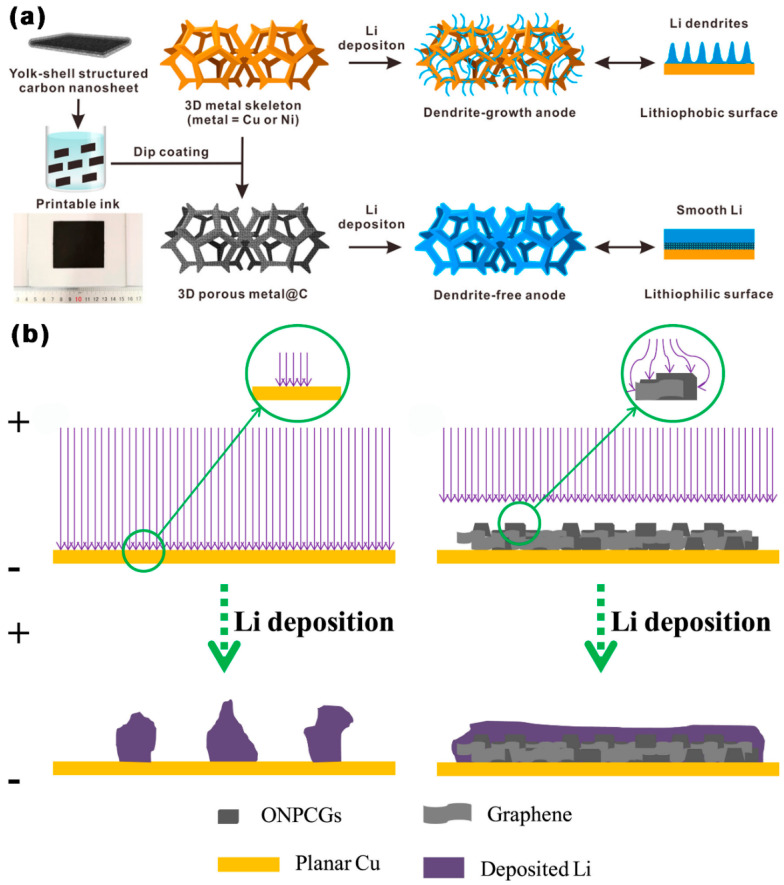
Schematic diagram of Li plating on (**a**) an NPCN-wrapped 3D metal foam [83], and (**b**) a Cu@ONPCG electrode [84].

**Figure 8 ijms-24-07291-f008:**
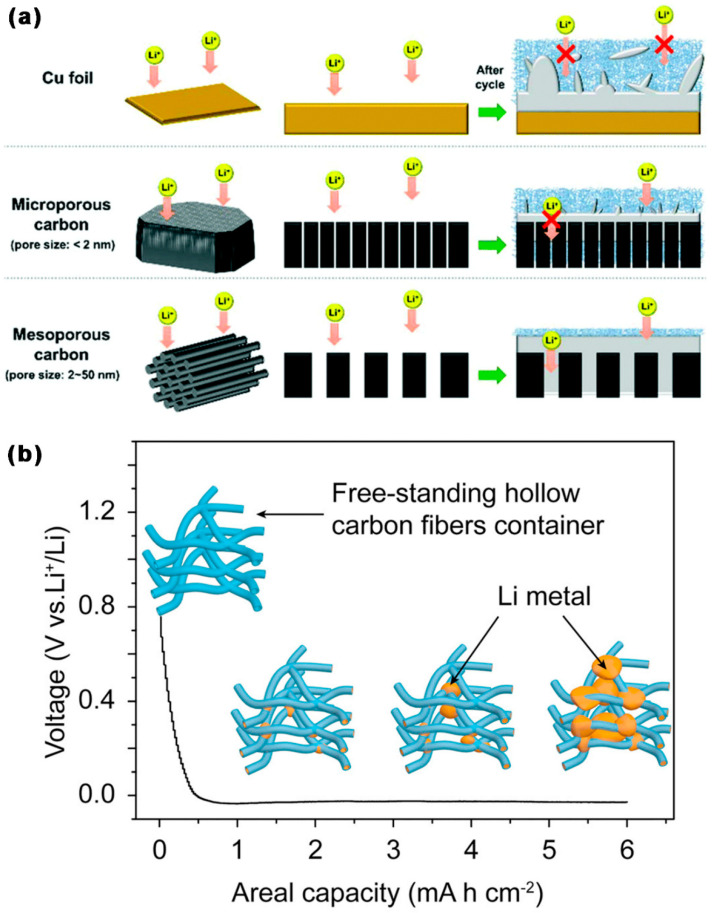
Schematic diagram of Li plating on (**a**) CMK-3 [85], (**b**) voltage profile of the lithium plating process and the corresponding schematic diagram [86].

**Figure 9 ijms-24-07291-f009:**
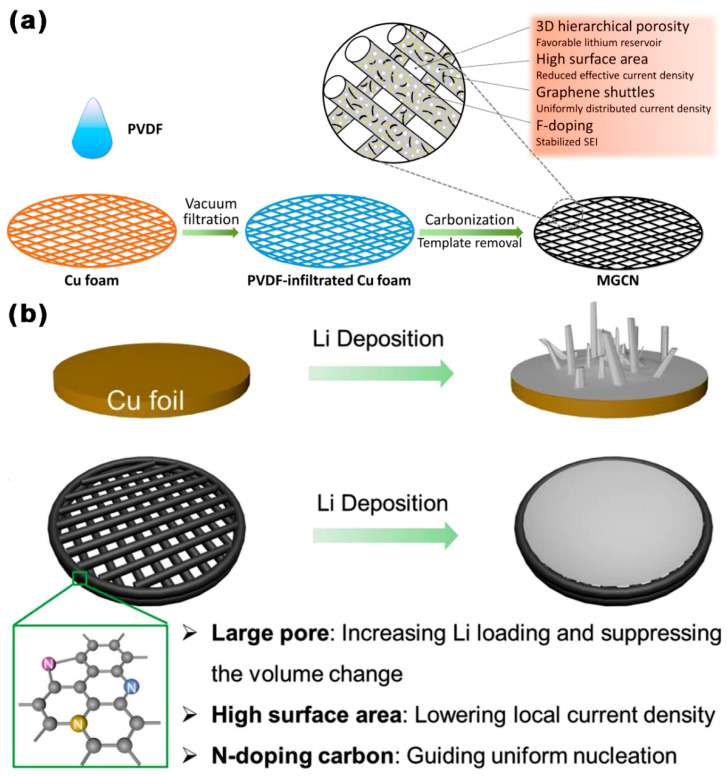
Schematic diagram of Li plating on (**a**) MGCN [87], and (**b**) a 3DP-NC electrode (yellow, purple, blue: N atoms) [33].

**Figure 10 ijms-24-07291-f010:**
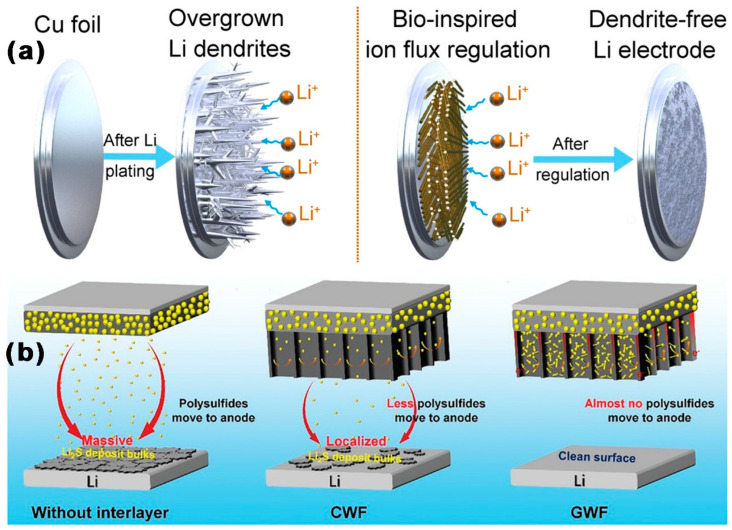
Schematic diagram of Li plating on Li-anode with (**a**) nanowood [88], and (**b**) a GWF interlayer electrode [89].

**Figure 11 ijms-24-07291-f011:**
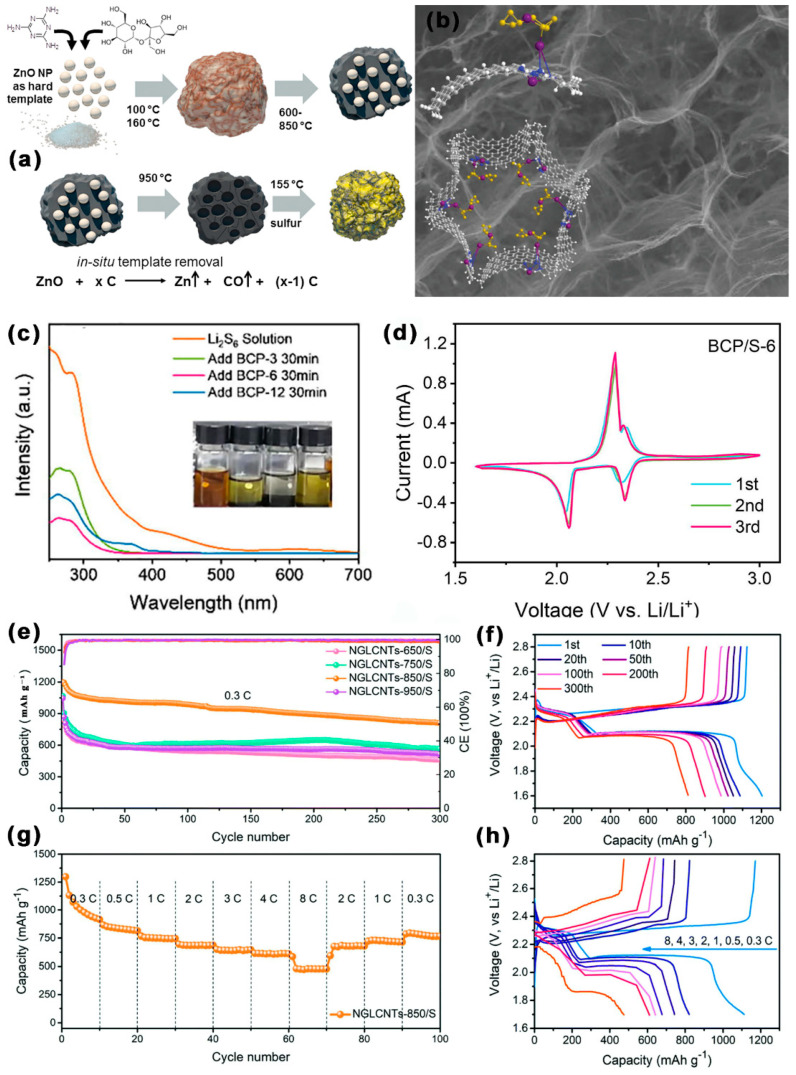
(**a**) Synthesis scheme of the hierarchical porous carbons (HPC) [96]. (**b**) SEM image of rNGO/S (the inset was a schematic) [97]. (**c**) UV–Vis absorption spectra (the inset was an optical photograph) of Li_2_S_6_ solution before and after adding BCPs. (**d**) CV curves of BCP/S-6 at 0.1 mV s^−1^. Electrochemical performances of NGLCNTs-850/S cathode [94]. (**e**) Cycling capacity and coulombic efficiency of NGLCNTs-850/S at 0.3 C. (**f**) Galvanostatic discharge and charge curves at 1–300 cycles. (**g**) Rate performance from 0.3 C to 8 C. (**h**) Galvanostatic discharge–charge curves at different C-rates [98].

**Figure 12 ijms-24-07291-f012:**
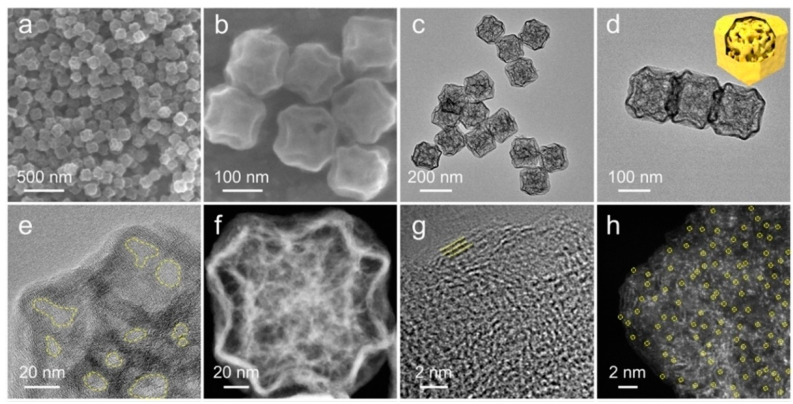
(**a**,**b**) SEM, (**c**–**e**) TEM, (**f**,**g**) HRTEM and (**h**) HAADF-STEM images of Ni-NC(p) (the inset in (**d**) is a schematic diagram of the structure of Ni-NC(p), the markers in (**g**) represent the thin shells of carbon, and the markers in (**h**) represent uniformly dispersed Ni single atoms) [48]. (**i**) Synthesis scheme of the three-dimensional (3D) conductive carbon foams (CFs) [100]. (**j**) Impedance graph of symmetric cells SC-Co electrodes. (**k**) Potentiostatic nucleation curves with SC-Co cathode. (**l**) Cycling performances of LSBs. (**m**) Charging/discharging profiles at different rates. (**n**) Cycling performances under the circumstance of a high S loading cathode up to 3.6 mg cm^−2^ [45].

**Figure 13 ijms-24-07291-f013:**
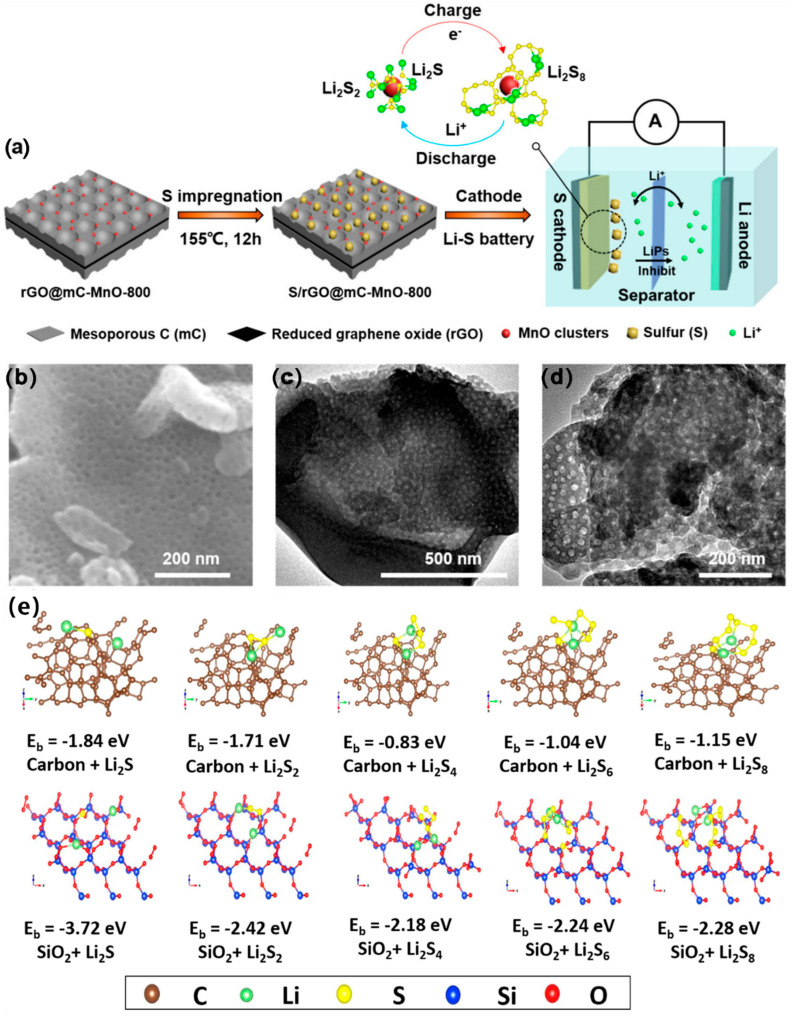
(**a**) Schematic diagram of the preparation of S/rGO@mC-MnO-800 composite electrode. (**b**) SEM and TEM (**c**,**d**) images of S/rGO@mC-MnO-800 composites [103]. (**e**) Theoretical DFT simulation of the interactions between SMC/SiO_2_ and LiPSs [102].

**Figure 14 ijms-24-07291-f014:**
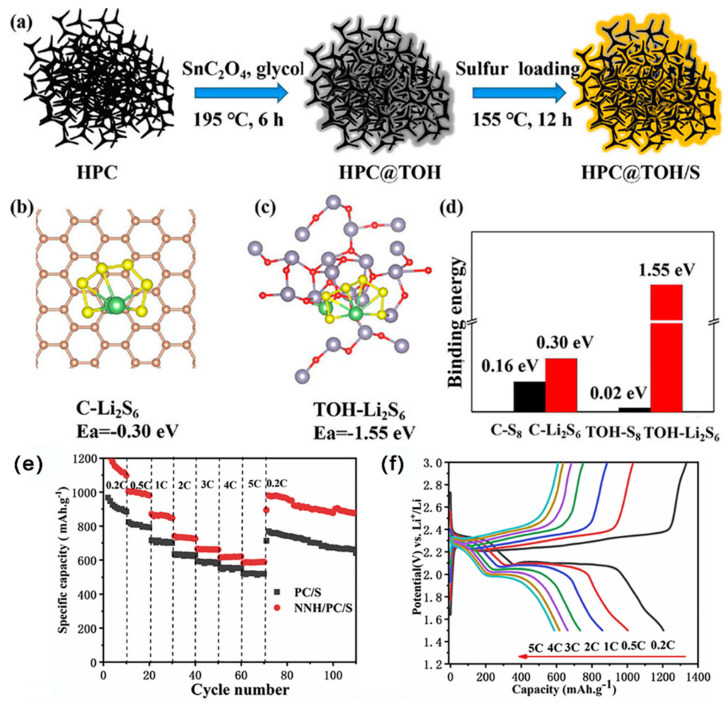
(**a**) Schematic diagram of the preparation of HPC@TOH/S, density functional theory calculations of the adsorption energy for Li_2_S_6_. Optimized atomic structures of (**b**) CLi_2_S_6_, (**c**) TOH-Li_2_S_6_, (**d**) comparison of the binding energies of Li_2_S_6_/S_8_ to C or TOH [104]. (**e**) Rate capability of LSBs with NNH/PC/S cathodes. (**f**) NNH/PC/S’ galvanostatic charge and discharge curves with varying current rates between 0.2 C and 5 C. (**g**) Long-term cycling performance of LSBs with NNH/PC/S cathodes at 0.5 C [105].

**Figure 15 ijms-24-07291-f015:**
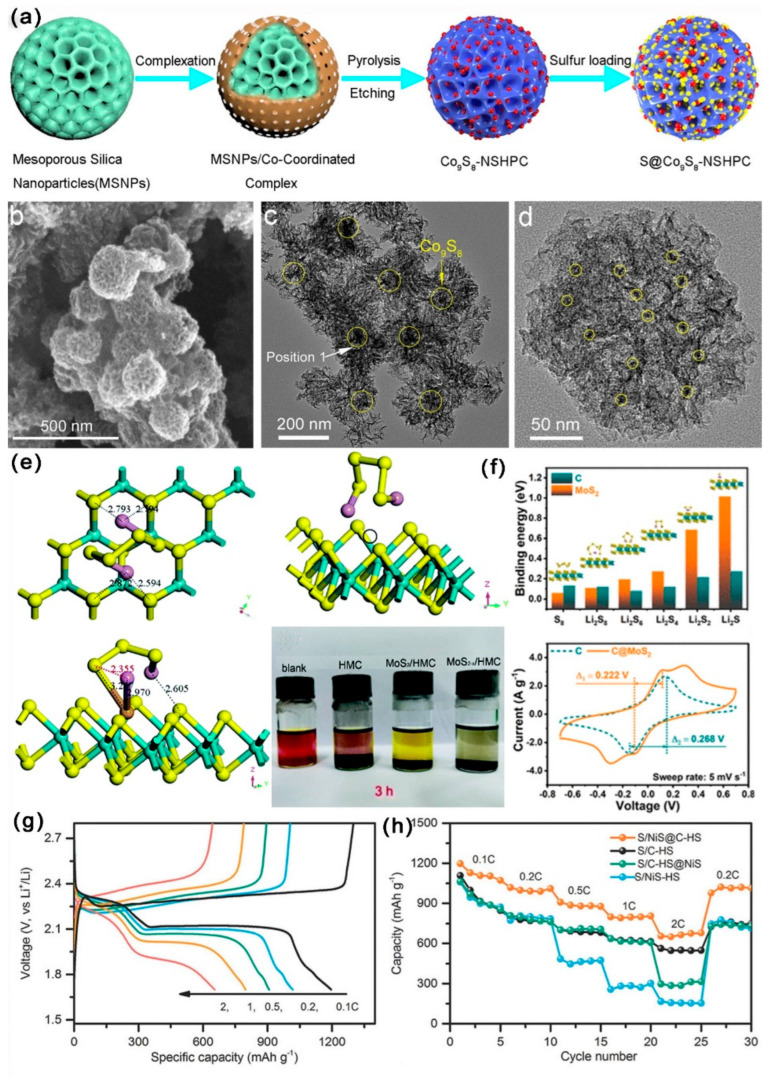
(**a**) Schematic diagram of the preparation of S@Co9S8-NSHPC, (**b**) SEM, (**c**,**d**) TEM images of Co_9_S_8_-NSHPC [106]. (**e**) Adsorption configurations of Li_2_S_4_ on MoS_2−X_ and visual adsorption experiments on a Li_2_S_6_ electrolytes after adding MoS_2−X_/HMC (yellow: S, blue: Mo, purple: Li, black circle: Mo–S bond) [107]. (**f**) Binding energies for Li_x_S_n_ species on MoS_2_ and CV curves of C@MoS_2_ electrode with 0.1 M Li_2_S_6_ [108]. (**g**) Discharge/charge curves and rate capabilities of S/NiS@C-HS. (**h**) Rate capabilities of S/NiS@C-HS at 0.2 C. [109].

**Figure 16 ijms-24-07291-f016:**
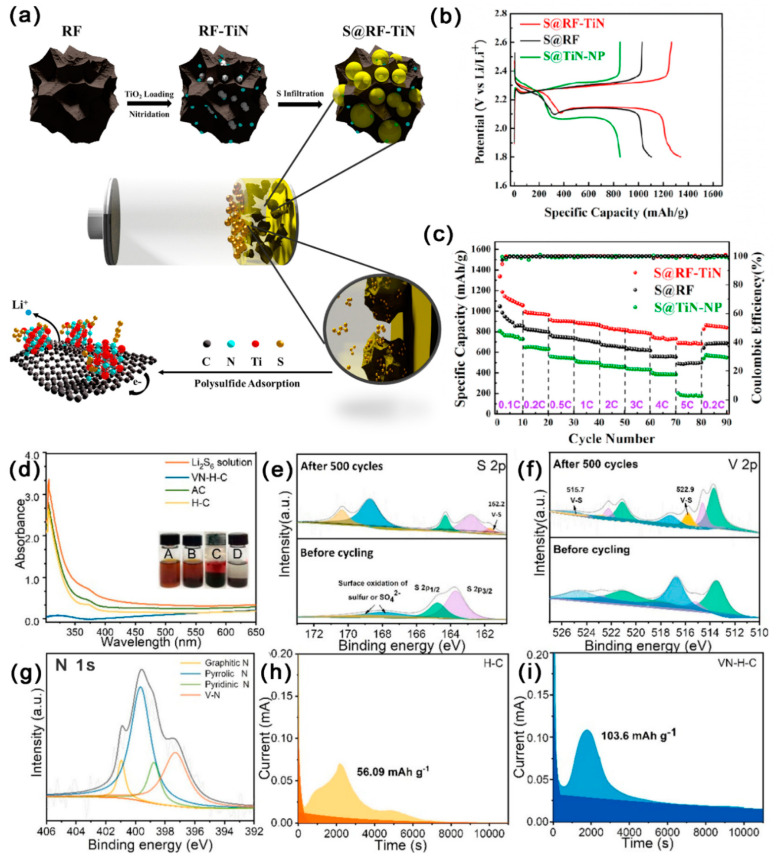
(**a**) Schematic diagram of the preparation of S@RF-TiN. (**b**) galvanostatic charge and discharge potential profiles of S@RF-TiN at 0.1 C. (**c**) Rate performance for S@RF-TiN (0.1 C–5 C) [110]. (**d**) Visual adsorption experiments on a Li_2_S_6_ electrolytes after adding MoS_2−X_/HMC, (**e**) S 2p, (**f**) V 2p and (**g**) N 1s XPS spectra of S@VN-H-C before and after 500 cycles. (**h**,**i**) Li_2_S_8_ and tetraglyme solution potentiostatic discharge patterns on VN-H-C surfaces at 2.05 V (The upper light-colored region represents the deposition of Li_2_S) [111]. (**j**) Diagram demonstrating the conversion and adsorption of LiPSs on the TiN [112].

**Figure 17 ijms-24-07291-f017:**
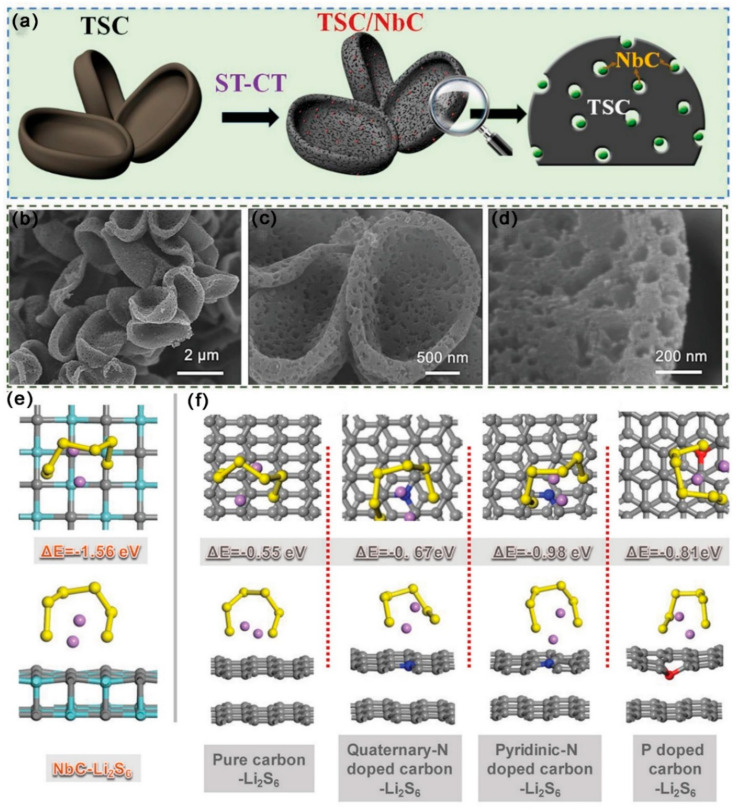
(**a**) Schematic diagram of the preparation of TSC/NbC. (**b**–**d**) SEM images of TSC/NbC, density functional theory calculations of the adsorption energy for Li_2_S_6_. Optimized atomic structures of (**e**) NbC-Li_2_S_6_, (**f**) pure C-Li_2_S_6_, quaternary-N/C-Li_2_S_6_, pyridinic-N/C-Li_2_S_6_ and P/C-Li_2_S_6_ [113]. (**g**) TGA curves under N_2_ atmosphere, (**h**) XRD patterns, (**i**) N_2_ adsorption/desorption isotherms (The upper curve represents the desorption isotherm, while the lower curve represents the adsorption isotherm), and (**j**) pore size distributions of HCN@NbC. (**k**) Mechanism of the action of Nb_2_O_5_/C and NbC/C on LiPSs [115].

**Figure 18 ijms-24-07291-f018:**
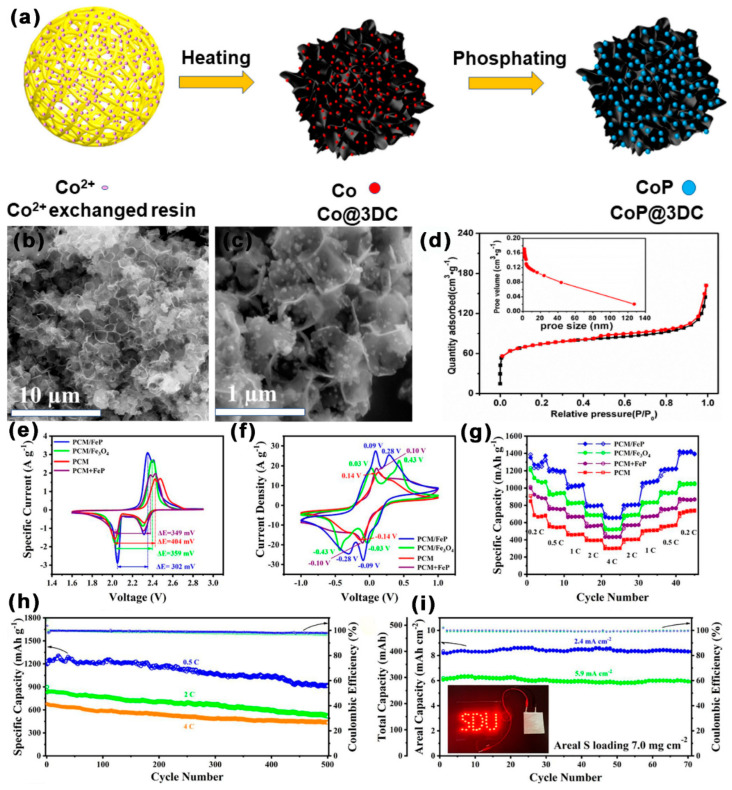
(**a**) Schematic diagram of the preparation of CoP@3DSC, (**b**,**c**) SEM images, (**d**) N_2_ adsorption/desorption isotherms of CoP@3DSC [116] (The upper curve represents the desorption isotherm, while the lower curve represents the adsorption isotherm). (**e**) CV profiles of LSBs with PCM/FeP/S cathode. (**f**) CV profiles of symmetric batteries. (**g**) Rating performance of LSBs with PCM/FeP/S cathode (0.2 C–4 C). (**h**) Long cycle of LSBs with PCM/FeP/S cathode. (**i**) Long cycle of pouch cell with PCM/FeP/S cathode (the inset shows an optical picture of a pouch cell in operation) [117]. 3C etc. is indicated above the solid line in (**h**) and 2.4 mA cm^−2^ etc. is marked above the dashed line in (**i**).

**Figure 19 ijms-24-07291-f019:**
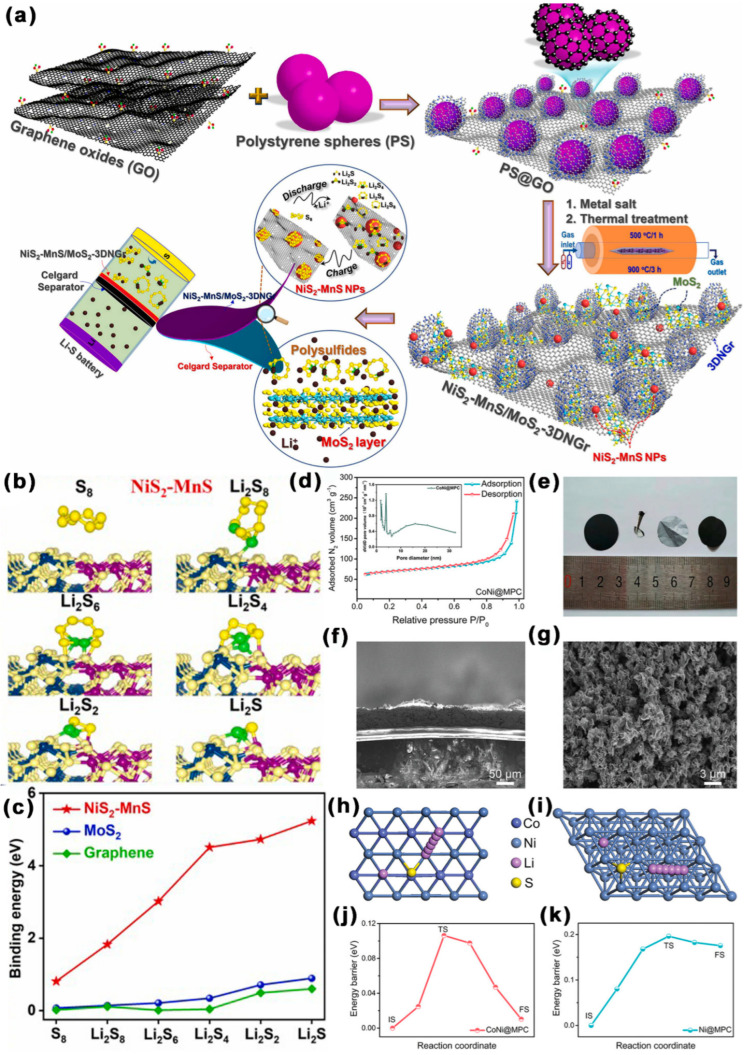
(**a**) Schematic diagram of the preparation of NiS_2_-MnS/MoS_2_-3DNGr. (**b**) Optimized atomic structures of NiS_2_-MnS with S species. (**c**) Binding energies for the S species and NiS-MnS [145]. (**d**) N_2_ adsorption/desorption isotherm of CoNi@MPC, (**e**) Optical photograph of the CoNi@MPC-optimized separator before and after folding. (**f**,**g**) SEM images of the CoNi@MPC-optimized separator. Decomposition pathways of Li_2_S on (**h**) CoNi (111) and (**i**) Ni (111); energy curve of Li_2_S disintegration on (**j**) CoNi (111) and (**k**) Ni (111) [146].

**Figure 20 ijms-24-07291-f020:**
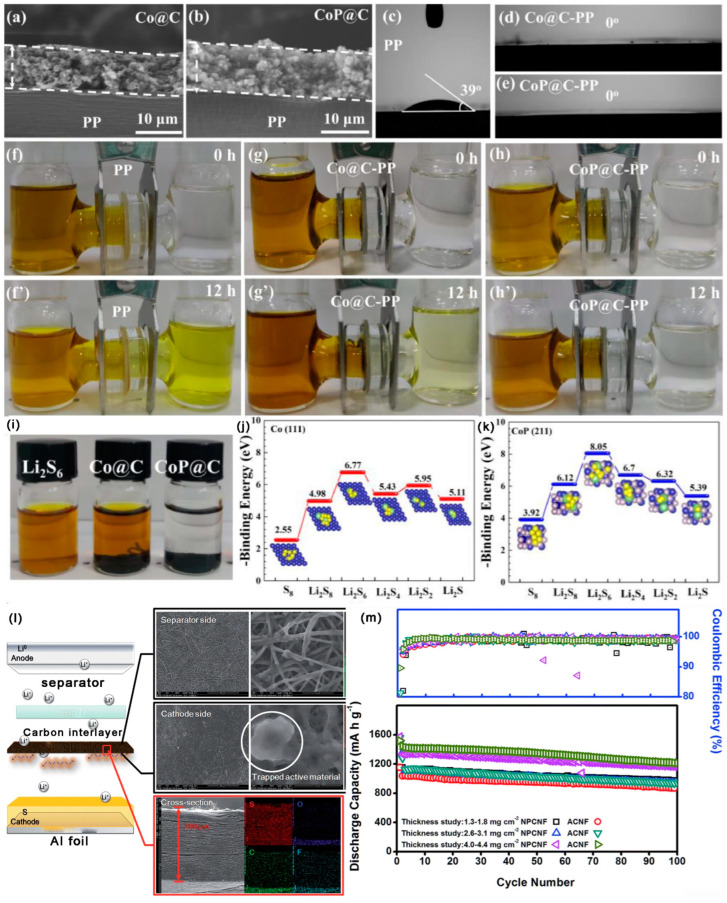
SEM images of separators with (**a**) Co@C coating and (**b**) CoP@C coating. Contact angles of separators with (**c**) blank, (**d**) Co@C coating, and (**e**) CoP@C coating. Experiments on H-shaped permeation devices of separators with (**f**,**f**’) blank, (**g**,**g**’) Co@C coating, and (**h**,**h**’) CoP@C coating. (**i**) The optical images of the CoP@C-containing Li_2_S_6_ solution adsorption experiments. The adsorption energies of the S-containing substance on (**j**) Co (111) and (**k**) CoP (211) [148]. (**l**) SEM and EDS mapping showed how the ACNF interlayer captured LiPSs. (**m**) Cycling performance of LSBs with ACNF interlayer [89].

**Figure 21 ijms-24-07291-f021:**
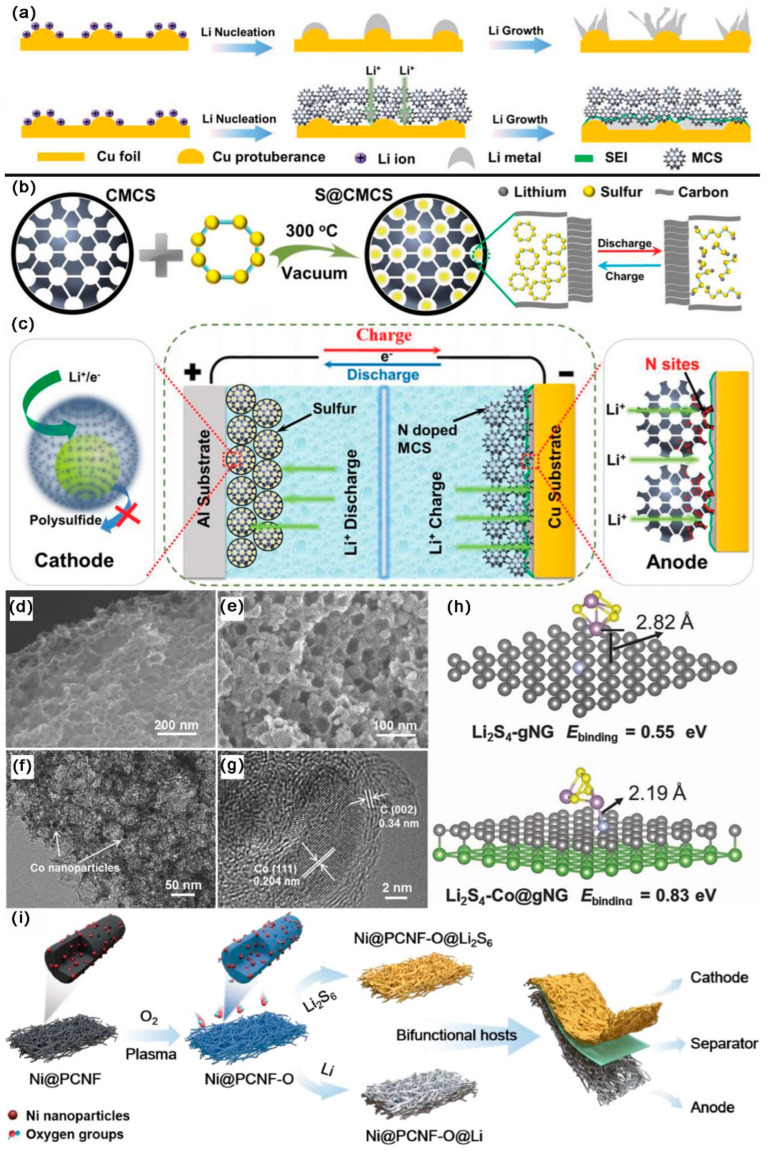
(**a**) Schematic diagram of Li growing on Cu foil optimized by MCS900. (**b**) Schematic diagram of the mechanisms of Ni@PCNF-O@Li_2_S_6_ during the battery operation process. (**c**) Schematic diagram of S@CMCS1100||Li@MCS900 batteries [150]. (**d**,**e**) SEM and (**f**,**g**) TEM images of Co/N-PCNSs; (**h**) Li_2_S_4_ and Co@gNG’s (the doped graphene with the introduction of graphitic N) binding energy was calculated using DFT (yellow: S, purple: Li, grey: C, green: Co) [151]. (**i**) Schematic diagram of the synthesis of Ni@PCNF-O@Li_2_S_6_ and Ni@PCNF-O@Li [152].

**Figure 22 ijms-24-07291-f022:**
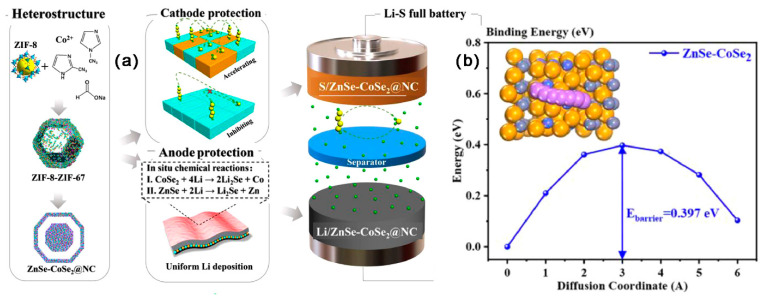
(**a**) The mechanism of S/ZnSe-CoSe_2_@NC||Li/ZnSe-CoSe_2_@NC. (**b**) Binding energy of Li_2_S_6_ on ZnSe-CoSe_2_. In situ growth experiments of lithium dendrites of symmetric batteries with (**c**) blank and (**d**) Li/ZnSe-CoSe_2_@NC [153]. (**e**) SEM images of separator with Ni_2_P-HCS-coated after cycling. (**f**) Long cycle of Li-S pouch battery with Ni_2_P-HCS-coated separator at 0.1 C. The inset shows a lithium-sulfur soft pack battery used to light up a LED light [154]. (**g**) f-BNNSs/f-CNTs’ optimal adsorption construction with LiPSs (yellow: S, green: Li, grey: C, blue: B, cyan: N, red: O, white: H). (**h**) Binding energies between S-containing substances and f-BNNSs/f-CNTs [155].

**Table 1 ijms-24-07291-t001:** A summary of the properties and LSBs performance of MCBM/S cathodes.

Materials	SSA (m^2^ g^−1^)	Mesoporous Diameter (nm)	TPV (cm^3^ g^−1^)	S Content (wt%)	S Loading (mg cm^−2^)	Initial DSC (mAh g^−1^)	Cycling Performance (mAh g^−1^)	CDR (%)	Rate Performance (mAh g^−1^)	Reference
MHCS/S	1875	~3.6	4.75	90.4	4.1	780/0.5 C	432/0.5 C 1100 cycles	0.054	476/2 C	[93]
rNGO/S	-	-	-	70	1.2	1186/0.1 C	837/0.1 C 200 cycles	0.147	997/1 C	[97]
HPC-N_2_	965	3.7	-	60	3	972/0.1 C	794/0.1 C 50 cycles	0.366	-	[96]
N-PC@uCo/S	1185.36	10–50	0.98	76	1.8	912/1 C	780/1 C 500 cycles	0.028	600/5 C	[118]
BCN@HCS/S	1057.9	2–4	0.72	70	4	1083/0.2 C	1041/0.2 C 50 cycles	0.038	670/3 C	[119]
NOMCs/S	1021	3.76	0.99	60.5	1.2–1.5	638.3/0.5 C	478.73/0.2 C 100 cycles	0.25	472.2/1 C	[120]
BCP/S-6	2032.2	2–4	1.03	80	1.5	1385/0.1 C	925/0.1 C 100 cycles	0.29	462/2 C	[94]
S-OMC-100S-2	1011.5	4.0	1.18	41	1	517/0.1 C	-	-	-	[95]
HMCS/S@GO	580	5.7	-	58.9	-	1054/0.5 C	635/0.5 C 100 cycles	0.398	626/2 C	[121]
NGLCNTs-850/S	142	2–45	-	71	1.0–1.5	1199.4/0.3 C	811/0.3 C 300 cycles	0.159	613/3 C	[98]
FBC/S	338.03	2.5	-	73.50	1.2	1145.9/0.1 C	1099.99/0.1 C 100 cycles	0.119	925.6/0.5 C	[122]
MPC/S	368.5	6	0.56	11.7	-	1584.56/250 mA g^−1^	804.94/250 mA g^−1^ 30 cycles	-	-	[91]
CMK-3	1976	3.33	2.1	70	1.82	1320/168 mA g^−1^	-	-	-	[35]
Ni-NC(p)/S	428.8	13	1.1	73.1	1.35–1.60	966.6/0.5 C	1600/0.1 C 1600 cycles	0.078	706.27/2 C	[48]
Co/PNC/S	588.0	~5	-	59.66	1.5	1105.4/0.5 C	746.7/0.5 C 100 cycles	0.324	540.6/1 C	[99]
CFs/S	156.8	9.8–14.7	-	70	3.69–3.71	855.6/135 mA g^−1^	586.5/135 mA g^−1^ 120 cycles	0.262	667.6/270 mA g^−1^	[100]
SC-Co/S	831	3–7	1.04	63	1.2	1130/0.5 C	837/0.5 C 300 cycles	0.086	-	[45]
Fe-N-C/S-MCF	1267	31	3.5	87.9	2.5	1631/0.5 C	1280/0.5 C 100 cycles	0.215	798/5 C	[123]
NMC-Al_2_O_3_/S	1485	12	2.25	73.5	2.0	902/0.5 C	685/0.5 C 1000 cycles	0.023	755/2 C	[101]
SMC/S	365.45	50	0.473	80	0.8~1.1	969.7/0.2 C	625.5/0.2 C 400 cycles	0.088	488.9/3 C	[102]
MnO-800/S	300.9	18	0.57	62	1.8	1535.9/0.2 A g^−1^	989.9/0.2 A g^−1^ 400 cycles	0.088	808/4 A g^−1^	[103]
NMC/La_2_O_3_/S	731	-	2.6	60	1.67	1043/1 C	799/1 C 100 cycles	0.234	475/5 C	[124]
GP/CNT/LNO-V-S	5	-	0.065	-	4.4	1007/0.2 C	962/0.2 C 100 cycles	0.045	844/1 C	[125]
S@C@MnO_2_/S	1087.1	5	1.96	58.2	3	983/1 C	550/1 C 500 cycles	0.088	465/5 C	[126]
MnO_2_@HCB/S	257	2–10	0.52	67.9	0.7–1	503/3 A g^−1^	-	-	496/4 A g^−1^	[127]
SiOx-coated CMK-3/S	-	3.5	-		1.2	718/0.1 C	592.3/0.1 C 60 cycles	0.292	897.4/1 C	[128]
MCS-SiO_2_/MXene/S	315.436	3.442	0.073	68	3.2	755.1/1 C	537.6/1 C 500 cycles	0.046	575.9/2 C	[129]
C-S-TiO_2_	-	-	-	53	-	1128/0.2 C	608/0.2 C 120 cycles	0.384	650/5 C	[130]
TiN-O-OMC/S	355.9	3.5	0.95	75	1.4	790/0.2 C	634/0.2 C 120 cycles	0.16	550/2 C	[131]
KC/S-60	2000	4.2–18	3	72	1.5	1115/0.1 C	820/0.1 C 120 cycles	0.33	-	[132]
TiO_2_/G/NPCFs/S	341.5	3.6	0.309	55	1.2	987/1 C	618/1 C 500 cycles	0.074	668/5 C	[133]
RuO_2_-MPC-HS/S	343	0.7–12.5	1.69	70	-	859/0.2 C	-	0.052	665/0.5 C	[134]
ZCO-QDs@HCS/S	782.50	5.0–9.5	-	70	1.3	1009.3/1 C	675.2/1 C 400 cycles	0.083	725.1/3 C	[135]
MCM/Nb_2_O_5_/S	948	10	2.6	60	1.5	1289/0.5 C	928.08/0.5 C 200 cycles	0.14	640/5 C	[136]
OMCNS	386.7	15–20	1.05	70	1–1.2	840	505.7/0.5 C 500 cycles	0.081	580.6/2 C	[137]
HPC@TOH/S	132.99	-	-	73.8	-	918.05/1 C	697.72/1 C 400 cycles	0.06	770.61/2 C	[104]
NNH/PC/S	2615	1–50	-	53.07	1.56	1203.5/0.5 C	521.3/0.5 C 700 cycles	0.081	583.9/5 C	[105]
Co_9_S_8_-NSHPC/S	521.42	2–30	-	75	1.2	918/0.2 C	867/0.2 C 200 cycles	0.028	607/2 C	[106]
MoS_2−X_/HMC/S	146.6	-	1.31	60	1.2	1077/0.2 C	754/0.2 C 100 cycles	0.3	528.3/5 C	[107]
C@MoS_2_/S	455.9	3–4	0.48	80	1.3	752.5/2 C	500/2 C 1000 cycles	0.03	554.2/5 C	[108]
NMCS@MoS_2_/S	323.5	13.02	1.14	70	1.2	847/1 C	813/1 C 500 cycles	0.08	-	[138]
C-HS@NiS	59	-	-	72	2.3	723.2/0.5 C	695/0.5 C 300 cycles	0.013	674/2 C	[109]
MoS_2_/CNT/S	180.2	-	0.9	60	2.6	1470/0.2 C	855.5/0.2 C 50 cycles	0.83	1254/5 C	[139]
MHCS@MoS_2_/S	742	40	1.026	72.1	1.5	980.93/1 C	735.7/1 C 500 cycles	0.05	886.0/2 C	[140]
RF-TiN/S	900	42	4.12	70	1.5	924/1 C	700/1 C 800 cycles	0.04	690/5 C	[110]
VN-H-C/S	316.42	5–20	-	74	1.5	856.5/1 C	602.5/1 C 500 cycles	0.059	789/2 C	[111]
C@TiN/S	277	-	-	71	1.1	457.58/3 C	453/3 C 300 cycles	0.0033	373/5 C	[112]
PCF/VN/S	-	-	-	60.1	8.1	1310.8/0.1 C	1052.5/0.1 C 250 cycles	0.07	591.6/5 C	[141]
TSC/NbC/S	555	30–50	-	66.02	2	1153.63/0.1 C	937.9/0.1 C 500 cycles	0.037	499/5 C	[113]
MHCS@Mo_2_ C/C/S	813.6	3–9	1.17	71.8	1.2	1316.8/0.1 C	880.1/0.1 C 100 cycles	0.33	678.8/2 C	[114]
HCN@NbC/S	1348	8	3.13	80	1.5	953/1 C	595/1 C 800 cycles	0.05	752/5 C	[115]
TiC/C/S	760	-	0.799	74	1.1	821/0.2 A g^−1^	602/0.2 A g^−1^ 200 cycles	0.133	438/2 A g^−1^	[142]
CoP@3DSC/S	867.78	5–15	-	73.3	1.17	1117.37/0.5 C	740.56/0.5 C 600 cycles	0.056	820.72/2 C	[116]
PCM/FeP/S	500	-	0.9	75	2	1231/0.5 C	910.7/0.5 C 500 cycles	0.05	655/3 C	[117]
Co/CoP@NC/S	20	-	-	75.8	2.5	848.54/1 C	638/1 C 1000 cycles	0.033	472/20C	[143]
CoFeP@CN/S	186	-	0.85	70	4.1	683.39/1 C	608/1 C 400 cycles	0.031	630/5 C	[144]

## Data Availability

Not applicable.

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
