# Peer review of "Mesoporous Carbon-Based Materials for Enhancing the Performance of Lithium-Sulfur Batteries"

_ijms, 2023, doi:10.3390/ijms24087291_

Round 1

Reviewer 1 Report

The paper looks to narrate the various synthesis procedure, but very little effort on comparing each other, merits and demerits. Efforts are made in the conclusion, which needs to be moved the main content.  Conclusion should be made short, clearly highlighting the state of art. 

Reviewer 2 Report

Overview:

The review ijms-2263877 reports on the progress made in the field of the electrodes of lithium-sulfur batteries from ~2009 to the present day. Authors are focused on the mesoporous carbon-based cathodes, which are considered promising materials designed to overcome the problems associated with purely sulfur electrodes. The review is quite comprehensive, and it presents the outlook aimed at the overcoming of existing problems, and there is no doubt that manuscript is of interest for IJMS readers. However, there are some questions to be answered and many inaccuracies to be fixed.

Major comments:

1) I encourage authors to comment on the porosity aspect: why mesopores are of particular interest; what pore sizes are optimal for Li storage; are SSA values presented in the text measured by gas absorption method, and, if yes, do they accurately represent the Li storage capacity of the electrodes; do hierarchical pore structures provide better availability of the pores? In general, what are the trends in the porosity adjustment and control?

2) What electrolytes are typically used the for LSBs with MCBM cathodes? Is electrolyte choice important for the performance of the batteries?

3) In section 3.4, please define the concept of Two-in-One Hosts and comment on its benefits in comparison to more "traditional" electrodes.

Minor comments:

1) Line 11, is “secondary high energy density batteries” phrase formulated properly?

2) Line 38, “This reactive batteries are divided into four main processes”: what do you mean by “reactive batteries”? Also, “this”->“these”.

3) Lines 42, 385: what do you mean by “rate capacity”, is this a proper term?

4) As abstract is a separate entity, define SSA (Line 54) and MCBM (Line 66) abbreviations in the main text once more.

5) Line 93: by “ording”, did you mean “ordering”?

6) Lines 127-128, “the amphiphilic molecules that serve as templates”: are individual molecules applied as templates, or did you mean a material comprised from such molecules?

7) Lines 178-179: what do you mean by “reverse mesoporous C material”?

8) Line 154, define “TEM”. Line 187, define “OTAB” and “BTEB”. Line 225, define “EISA”. Line 238, define “LDH”. Line 250, define “TSIL”. Line 345, define “SEI”. The abundance of abbreviations sometimes interfere with the clarity of the text. Authors are encouraged to create a list of abbreviations and to reduce their usage whenever it is possible.

9) Fig. 3(c), fig.4(a,b,e,f,g), fig. 5(a,b) – please indicate if TEM or SEM images are presented.

10) Line 215: “network[66]. (47) (e) SiO2”: what is (47)?

11) Line 252: revise “porous carbon had abundant mesoporous” fragment.

12) In line 259, by “TOF”, did you mean “TPV”?

13) Line 275, in “N/S-HMCS900”, what does “900” mean?

14) Line 289, revise “which decreased Li activity and passivates” fragment.

15) Line 297, “yolk shell structure” is more commonly referred to as “yolk-shell structure” or “yolk/shell structure”.

16) Please revise the caption of fig. 8(b) (line 337).

17) Line 349: “98% for 150 cycles [87]. (8) Wang et al. synthesized”, line 355: “for 75 cycles [33]. (9)”: what is (8) and (9)?

18) Line 352, revise the “accommodate a large amount of Li deposition” fragment.

19) Line 394-395 “Physical restrictions on polar LiPSs, however, can only be offered by the pore structure because carbon compounds are non-polar in nature.”: carbon can form polar compounds, for example, CO, or the ones based on C-F bonds. Please revise the statement.

20) Lines 255, 455: in “abundant mesoporous”, mesoporous => mesopores

21) Lines 126, 197, 231… “mesoporous pore” => “mesopore”

22) Fig. 12: please indicate in the caption what is shown in the inset of fig.12(d). What is marked in fig.12(g,h)? In fig.12(j), is half of the subfigure empty on purpose?

23) Line 430, “rNGO/S ( The illustration was a schematic) [97]. (c) UV–vis absorption spectra (The illustration was an optical photograph)”: by illustration, did you mean “inset”?

24) Is the caption of Fig. 11(e) (“(e) galvanostatic discharge and charge profiles at 0.3 C;”) correct (lines 433-434)?

25) Line 533, “DFT theoretical calculation of adsorption energy Li2S6 with (b) C, (c) TOH,”: did you mean “the results and schematics of… obtained by … calculation”? This is also relevant for the caption of fig. 17, lines 617-618.

26) Line 533: by “HPC@TOH/S, (b) DFT”, did you mean “HPC@TOH/S, (b-c) DFT”?

27) Fig. 15, line 565: where is a caption of the fig. 15(h)?

28) In Fig. 16(g), could you indicate the “N1s” notation, similarly to the notations shown in fig.16 (e,f)?

29) In the Fig. 16, e) is S2p, f) is V 2p, g) is N1s, therefore lines 588-589 should be revised.

30) Fig. 17: (b-d) indexes are not visible, please change their color or create white outlines.

31) Line 625, by “Composites metal phosphides” you meant “Composite metal phosphides” or “Composites based on metal phosphides”?

32) In Fig. 18(i), are the lines adapted from refs. 28-38 related to the references cited in this review? If they are not, I suggest they should be removed.

33) In fig. 18(l), what is shown in the inset?

34) Lines 689-690, revise the fragment “interface binding Optimized structure”.

35) In fig. 19(e), what is shown in the inset?

Reviewer 3 Report

-      The abstract is not succinctly and scientifically written. So many interesting portions of an abstract writing style are missing. Such as the impacts of the findings of the study on the case study area. The implication of the results to individual as well as the overall case study and to existing or future policy issues etc. Some few reasons behind the adoption of the methodology is also not visible in the present abstract. These are very necessary in order to entice the general audience to appreciate the quality of the study and motivate readers and other researchers to read through the article.-

-The motivation of the paper needs improvement. Why is the chosen methodology appropriate?

- The limitation of the study is not clear

-The study failed to discuss the policy impacts and implication, and the likely factors to be taken to mitigate most of the policy implications identified in the study. The study only repeated what was earlier discovered and same was done in the conclusion of the study.

Reviewer 4 Report

The review article "Mesoporous Carbon-Based Materials for Enhancing the Performance of Lithium-Sulfur Batteries" provides a comprehensive overview and nicely summarizes the challenges associated with Li-S batteries. In particular, the shuttle effects, volume change, and poor electron and ion conductivity of Li2S2 and Li2S phases. The role of using mesoporous carbon-based materials to alleviate these issues has been highlighted in the article. The overall quality and the structure of the review article are great and would be of importance to the readership of the International Journal of Molecular Sciences.

I would accept this article in its present form.

Author Response

Dear reviewer,

     At the outset, we are thankful to the reviewer for your carefully reading  and valuable comments, your recognition of our research is also greatly appreciated.

Round 2

Reviewer 2 Report

Authors have addressed my questions and significantly improved the manuscript. However, some minor points should still be revised.

1) Line 187-188, “mesoporous C material with the opposite structure to that of the hard film plate MCM-4”: I still don’t quite understand what is an “opposite structure”. Please describe this concept in more detail.

2) Line 408: “for 200 cycles [89]. (9) Guo et al.”: I suggest that (9) should be removed.

3) Line 420: “every cycle [90]. (11)”: “(11) should be removed.

4) Fig. 21 and line 825: as I understand, the “gNG” abbreviation i4) s not defined in the text. Please revise.

6) Fig. 22(f): please indicate what is shown in the inset.

7) By “which can accommodate Li and S and mitigate their volume variation” (line 783), did you mean that the materials can mitigate the variation of the electrode volume induced by Li/S intercalation? Word “their” looks misleading to me.

Reviewer 3 Report

The authors revised the paper based on my comments 

Author Response

Thanks for your constructive and insightful comments again.